



# A new exponentially-decaying error correlation model for assimilating OCO-2 column-average $CO_2$ data, using a length scale computed from airborne lidar measurements

David F. Baker[1], Emily Bell[1], Kenneth J. Davis[2,3], Joel F. Campbell[4], Bing Lin[4], and Jeremy Dobler[5]

[1]Cooperative Institute for Research in the Atmosphere, Colorado State University, Fort Collins, CO, USA
[2]Department of Meteorology and Atmospheric Science, Pennsylvania State University, University Park, PA, USA
[3]Earth and Environmental Systems Institute, Pennsylvania State University, University Park, PA, USA
[4]NASA Langley Research Center, Hampton, VA, USA
[5]Spectral Sensor Solutions LLC, Fort Wayne, IN, USA

**Correspondence:** David Baker (david.f.baker@noaa.gov)

**Abstract.** To check the accuracy of column-average dry air $CO_2$ mole fractions ("$X_{CO_2}$") retrieved from Orbiting Carbon Overvatory (OCO-2) data, a similar quantity has been measured from the Multi-functional Fiber Laser Lidar (MFLL) aboard aircraft flying underneath OCO-2 as part of the Atmospheric Carbon and Transport (ACT)-America flight campaigns. Here we do a lagged correlation analysis of these MFLL–OCO-2 column $CO_2$ differences and find that their correlation spectrum falls

off rapidly at along-track separation distances of under 10 km, with a correlation length scale of about 10 km, and less rapidly at longer separation distances, with a correlation length scale of about 20 km.

The OCO-2 satellite takes many $CO_2$ measurements with small ($\sim$3 km$^2$) fields of view (FOVs) in a thin (<10 km wide) swath running parallel to its orbit: up to 24 separate FOVs may be obtained per second (across a $\sim$6.75 km distance on the ground), though clouds, aerosols, and other factors cause considerable data dropout. Errors in the $CO_2$ retrieval method have

long been thought to be correlated at these fine scales, and methods to account for these when assimilating these data into top-down atmospheric $CO_2$ flux inversions have been developed. A common approach has been to average the data at coarser scales (*e.g.*, in 10-second-long bins) along-track, then assign an uncertainty to the averaged value that accounts for the error correlations. Here we outline the methods used up to now for computing these 10-second averages and their uncertainties, including the constant-correlation-with-distance error model currently being used to summarize the OCO-2 version 9 $X_{CO_2}$

retrievals as part of the OCO-2 flux inversion model intercomparison project. We then derive a new one-dimensional error model using correlations that decay exponentially with separation distance, apply this model to the OCO-2 data using the correlation length scales derived from the MFLL–OCO-2 differences, and compare the results (for both the average and its uncertainty) to those given by the current constant-correlation error model. To implement this new model, the data are averaged first across 2-second spans, to collapse the cross-track distribution of the real data onto the 1-D path assumed by the new model.

A small percentage of the data that cause non-physical negative averaging weights in the model are thrown out. The correlation lengths over the ocean, which the land-based MFLL data do not clarify, are assumed to be twice those over the land.

The new correlation model gives 10-second $X_{CO_2}$ averages that are only a few tenths of a ppm different from the constant-correlation model. Over land, the uncertainties in the mean are also similar, suggesting that the +0.3 constant correlation





coefficient currently used in the model there is accurate. Over the oceans, the twice-the-land correlation lengths that we assume

here result in a significantly lower uncertainty on the mean than the +0.6 constant correlation currently gives – measurements similar to the MFLL ones are needed over the oceans to do better. Finally, we show how our 1-D exponential error correlation model may be used to account for correlations in those inversion methods that choose to assimilate each $X_{CO_2}$ retrieval individually, and to account for correlations *between* separate 10-second averages when these are assimilated instead.

## 1 Introduction

Column-averaged $CO_2$ mixing ratio measurements taken from satellites provide coverage across the globe that is far more extensive than that from *in situ* measurements. These satellite measurements are often used in global atmospheric flux inversions to provide a "top-down" constraint on surface sources and sinks of $CO_2$. The atmospheric transport models underlying these global inversions are generally run at a coarse resolution, using grid boxes of 100s of km on a side. The resolution is limited both for computational reasons (the models must be run many dozens of times across the measurements to obtain the inverse

estimate) and because the spatial coverage of the satellite measurements is currently not dense enough to resolve spatial scales much finer than this when solving at typical time scales (the gap in longitude between subsequent passes of a typical low-Earth-orbiting (LEO) satellite is $\sim25°$, resulting in gaps of between $3°$ and $4°$ across a week, gaps which are generally never filled in further due to the repeat cycle of the satellite's orbit). The typical field-of-view (FOV) of individual retrievals is often much smaller than this gridbox scale, however: FOVs for retrievals from the Orbiting Carbon Observatory (OCO-2) satellite (Crisp et

al., 2004; 2008), for example, are typically $\sim2.25$ km along-track by at most 1.25 km across-track (Eldering et al., 2017). The individual OCO-2 retrievals are generally averaged together along-track across some distance closer to the model grid box size before being assimilated in the inversion: this is because the modeled measurements to which the true measurements will be compared in the inversion are available only at the grid box resolution, so it makes little sense to assimilate each measurement individually when assimilating a coarse-resolution summary value will do just as well.

Whether the individual OCO-2 retrievals or some coarser-resolution average measurement are assimilated into the inverse model, correlations between the errors in the individual $CO_2$ measurements must be considered. $CO_2$ mixing ratios in the upper part of the atmospheric column (at all levels but the immediate surface layer) feel the influence of multiple flux locations at the surface due to atmospheric mixing, causing errors in adjacent measurements to be highly correlated there. Also, systematic errors in the individual $CO_2$ retrievals are correlated at finer scales because incorrect assumptions are made about the scatterers,

water vapor, temperature, and surface properties used in the retrieval scheme, and these variables themselves have errors that are correlated at these scales. The OCO-2 satellite, for example, makes 24 separate observations per second, across a distance of $\sim6.75$ km along the ground track: these data provide mostly-redundant column $CO_2$ information across that time. When deciding how to weight the satellite data in the inversions relative to the *a priori* information, some assumptions about these measurement error correlations must therefore be made: if these errors were assumed to be all independent, the total amount of

measurement information going into the inversions would be much too high, resulting in improper weighting *versus* the *a priori* or dynamically-propagated information in the problem. Our goal here is to present a new model of the errors in the OCO-2 $CO_2$





measurements that assumes correlations that die off exponentially with distance, as opposed to the constant-correlation models used previously. This new model will allow the $CO_2$ measurements to be weighted more accurately in inversions, yielding more accurate $CO_2$ flux estimates.

Until recently, there have not been any good ground-truth data available to evaluate the satellite $CO_2$ measurement errors at finer scales. At the coarser scales, data from the Total Carbon Column Observing Network (TCCON) have been used to assess the magnitude and seasonal variability of OCO-2 errors (Wunch et al., 2017), as well as what portion of these might be considered random as opposed to systematic (Kulawik et al., 2019). Worden et al. (2017) have done a similar random versus systematic partitioning of OCO-2 errors by looking at the variability of retrieved $X_{CO_2}$ across small areas inside of which the

real $CO_2$ is thought not to vary much. Comparison to realistic $CO_2$ fields given by atmospheric transport models (e.g. using plausible prior flux estimates and forced to agree with the available *in situ* $CO_2$ measurements) have also been used to assess systematic errors in the satellite retrievals (O'Dell et al., 2012; 2018) at seasonal time scales and regional spacial scales. The TCCON sites provide column-averaged $CO_2$ measurements that can be compared directly to the OCO-2 column measurements, but because they are available only in a few fixed locations, they cannot assess how these errors vary along-track. There have

been many aircraft underflights of OCO-2, but these generally have taken only *in situ* measurements of $CO_2$, representative of a particular elevation rather than a column average. Some of these flights provide data across most of the atmospheric column (i.e., vertical profiles), but only generally at widely-spaced locations. None of these data are really well-suited for assessing along-track errors in the column average.

   Over the past several years, however, column-average measurements of $CO_2$ from aircraft-based lidars have become avail-

able. Several of these lidars were instrument test-beds developed as part of NASA's Active Sensing of $CO_2$ Emissions over Nights, Days, and Seasons (ASCENDS) satellite project (Jucks et al., 2015). One of these, the Multi-functional Fiber Laser Lidar (MFLL) (Dobbs et al., 2008; Dobler et al., 2013), has been flown (Campbell et al., 2020) as part of NASA's Atmospheric Carbon and Transport (ACT) - America project, an effort to detail $CO_2$ variability as a function of weather and front location across the eastern half of North America (Davis et al., submitted). Several of these MFLL flights were designed to pass un-

derneath OCO-2 along its ground track as it passed by, allowing $CO_2$ from the better part of the full column to be compared between the two. Bell et al. (2020) have lined up the MFLL and OCO-2 data for these flights as a function of horizontal location, and have assessed the accuracy of the along-track change in $CO_2$ (the linear slope) from OCO-2 using the MFLL data as a ground truth: they accounted for the differences in the vertical averaging kernels and vertical extent between the two measurements in doing this comparison. Here, we use this same Bell et al. (2020) dataset to assess the along-track correlation

length scale of the MFLL–OCO-2 measured column $CO_2$ differences.

   A team of inverse modelers using the OCO-2 $X_{CO_2}$ retrievals have formed an OCO-2 flux inversion model intercomparison project (MIP) to help differentiate the $CO_2$ fluxes robustly constrained by the OCO-2 data from the confounding biases in those same data: doing this as a MIP helps mitigate the impact of the errors in individual transport models (by looking at the results across the full ensemble of models) and the impact of any differences in the measurements and measurement errors

used in the inversions (all MIP participants were directed to use the same 10-second average OCO-2 $X_{CO_2}$ measurements and uncertainties). Here, we use the OCO-2 $X_{CO_2}$ 10-second averaging problem as an application to test the impact of the





newly-computed length scale. We interpret the MFLL–OCO-2 differences as OCO-2 retrieval errors and use the associated error correlation length scale to formulate a new model for the 10-second average $X_{CO_2}$ value and its uncertainty. We then compare the results of this new error model to the results of the current error model to assess the impact of the newly-calculated

correlation length scale, both on the absolute values of $X_{CO_2}$ as well as on the uncertainty calculated for the 10-second average values.

To better present the logical flow of our argument, we will split the paper into two parts, presenting both the method and results of our MFLL–OCO-2 analysis in the first (Sect. 2), then the method and results for the OCO-2 averaging application in the second. Since a few different averaging approaches have been used over time with the OCO-2 data, these previous methods

will be outlined in this second section for context, before describing a new averaging approach using the correlation length scale. Section 3.1 presents a general framework for the OCO-2 data averaging approach, leaving the form of the correlation matrix, **C**, general. Section 3.1.1 discusses averages in which all the averaged retrievals are assumed to be independent (i.e. a diagonal **C**), Sect. 3.1.2 the case in which the errors in all the averaged retrievals are assumed to be correlated with all the others in the span with the same positive correlation coefficient (all the off-diagonal terms in **C** having the same value), and Sect. 3.1.3

the case in which the correlations are assumed to die out exponentially with distance along the satellite track (exponential decay off main diagonal in **C**). Section 3.2 discusses a pathology in this approach for two of the error models – for some retrieval uncertainty value combinations, the weights given to some terms in the weighted average may go negative, violating a key criterion for any average – and gives a couple fallback options for handling it. Section 3.3 applies these different correlation models (including both those with the fallback weighting and those without it) to two simple example cases, calculating the

total measurement information content given by each model. Section 3.4 shows how the exponentially-decaying correlation model (or "exponential" model, hereafter) may be used to compute the effect of correlations *between* the full-span averages themselves (instead of the values going into them). In Sect. 3.5, we assess the impact of the correlation length scale determined from the MFLL–OCO-2 data on the averages of actual OCO-2 version 10 $X_{CO_2}$ retrievals, by calculating 10-second average values and their uncertainties using the new exponential correlation error model and comparing them to those given by the

constant-correlation error model. Finally, we discuss the implications of the new length-scale-dependent correlations in the Conclusion.

## 2   Computing a correlation length scale from measured MFLL–OCO-2 column-average CO$_2$ differences

### 2.1   MFLL measurements and their pairing with OCO-2 overflight data

The NASA-funded Atmospheric Carbon and Transport (ACT) - America project has flown five aircraft campaigns over the

2016-2019 time period, measuring CO$_2$, CH$_4$, and meteorological variables in an effort to understand the relationship between atmospheric carbon and weather patterns, as well as the processes driving the uptake and release of carbon. These campaigns were done across all four seasons, each with flights in the Mid-Atlantic, Mid-West, and Gulf Coast regions of North America (Davis et al., submitted). As part of this effort, flights were made along the ground tracks of the Orbiting Carbon Observatory





| Date | Data | Length | Altitude | Leg start | | Leg end | | Crossing | | Flight |
| [YYYYMMDD] | Points | [km] | [km] | °lon | °lat | °lon | °lat | °lon | °lat | location |
| --- | --- | --- | --- | --- | --- | --- | --- | --- | --- | --- |
| 20160727 | 46 | 414 | 9.0 | -77.6 | 39.5 | -78.8 | 43.1 | -78.0 | 40.8 | Pennsylvania |
| 20160805 | 59 | 539 | 9.0 | -102.1 | 43.3 | -103.7 | 48.0 | -103.3 | 46.8 | S. & N. Dakota |
| 20170215 | 58 | 535 | 8.7 | -101.1 | 40.1 | -99.4 | 35.5 | -100.3 | 38.2 | Kansas - Oklahoma |
| 20170308 | 43 | 453 | 6.0 | -76.9 | 37.0 | -78.1 | 40.9 | -77.2 | 38.0 | Virg.-Maryland-Penn. |
| 20171022 | 78 | 501 | 8.9 | -98.6 | 35.4 | -100.2 | 39.7 | -99.6 | 38.2 | Oklahoma - Kansas |
| 20171027 | 66 | 618 | 8.5 | -99.5 | 35.3 | -101.2 | 40.7 | -100.4 | 38.5 | Oklahoma - Kansas |

**Table 1.** Information on which MFLL–OCO-2 co-location points from which ACT-America flight legs were used in this study, including the crossing locations (points of closest approach between the aircraft and OCO-2 FOVs). See Bell (2018) and Bell et al. (2020) for more details.

(OCO-2) satellite, instrumented with downward-viewing lidars taking $CO_2$ measurements that could be compared with the

column-averaged $CO_2$ data taken by OCO-2.

The OCO-2 satellite measures radiances that are sensitive to dry air $CO_2$ mixing ratios throughout the depth of the atmospheric column. $CO_2$ mixing ratios are retrieved from these data at 20 levels in the vertical, evenly-spaced in terms of pressure. Because there is not enough information to robustly differentiate these 20 values, a single pressure-weighted vertical average value, $X_{CO_2}$, is computed from these. How much information is contributed to the $X_{CO_2}$ value from each level, as opposed to

being taken from some prior guess, is shown by the shape of the vertical averaging kernel vector (see Fig. 6 of Bell et al., 2020): a value of 1 meaning all from the measurement, of 0 meaning all from the prior. The satellite flies in a sun-synchronous orbit, passing over the equator at about 1:30 pm local time, as the Earth rotates underneath it; its $81°$ orbital inclination means that the flight path over North America is tilted somewhat, with the basic south-to-north motion going a bit SE-to-NW. Its 7077.7 km orbital semi-major axis results in a velocity of $\sim$6.75 km/s for its field-of-view (FOV) on the surface, and gives 15 or 16 orbits

per day. The observed path is at most 10 km wide, located along the ground track in nadir-viewing mode, and roughly parallel to it in glint mode: there are 8 FOVs in the cross-track direction, each at most 1.25 km wide, and three cross-track scans are taken per second, making each FOV extend $\sim$2.25 km in the along-track direction.

The lidar in question, the Multi-functional Fiber Laser Lidar (MFLL) (Dobbs et al., 2008; Dobler et al., 2013), was one of several flight instruments developed as a testbed for the $CO_2$ lidar to be used aboard NASA's proposed Active Sensing of $CO_2$

Emissions over Nights, Days, and Seasons (ASCENDS) satellite (Jucks et al., 2015). We examine MFLL data from six OCO-2 underflights here, four taken over the Great Plains, two over the Mid-Atlantic (Table 1). The flight legs were generally around 500 km in length, the aircraft taking about an hour to fly that distance. The satellite FOV, on the other hand, would take only about 75 seconds to traverse the same route, so that, although the aircraft and satellite would be looking at very close to the same point on the ground at some point during the flight, the time difference in viewing could be up to 40 min or so at the

ends of each leg: some change in the $CO_2$ actually measured at the same location could thus be expected due to the blowing winds. The lidar was carried on a C-130 aircraft flying generally 8-9 km above ground level, or about at 350 hPa; because





the OCO-2 measurements give lower weight to the upper parts of the column, the MFLL data are therefore able to provide an independent validation constraint on at least the lower 2/3 of the OCO-2 column averages. The vertical weighting of the two measurements, as embodied in their averaging kernel vectors, is also different, with the MFLL instrument giving more weight to the upper part of the measured column, just underneath the flight level, while the OCO-2 weight is more flat with pressure (see Fig. 6 of Bell et al., 2020). Lining up the two sets of measurements, and accounting for the differences in vertical weighting, has fortunately already been done by Bell (2018) and Bell et al. (2020). They used this data to test the accuracy of the spatial trend in $CO_2$ retrieved by OCO-2 across the flight legs; here, we will use a subset of that same data set to look at shorter-scale spatial variability (Baker et al., 2020). The reader is referred to Bell et al. (2020) for further details of the MFLL and OCO-2 measurements, as well as the comparison method and the details of the measurements on each flight leg.

## 2.2 Method for analyzing a correlation length scale

As described in Bell (2018) and Bell et al. (2020), the MFLL data have been binned and averaged across 60-second blocks, corresponding to swaths of from 7 to 9 km in length along the OCO-2 ground track, depending how fast the C-130 aircraft was flying at the time. All valid cloud-free OCO-2 retrievals (those declared "good" by the OCO-2 quality screening criteria) falling within the MFLL horizontal location range are similarly averaged. (Note, again, that the co-located MFLL and OCO-2 data may have measurement times that differ by up to an hour or more.) Only those spans with more than about 20 MFLL and 3 OCO-2 measurements are used in the analysis. Here we use the satellite FOV latitude and longitude to calculate the distance between different measurement blocks. We subtract the OCO-2 average from its corresponding MFLL average for each common bin, then divide the difference $X_j$ by the following OCO-2 $X_{CO_2}$ measurement uncertainty value:

$$\sigma_j^2 = \sigma_{j,retr}^2 + (0.3 \text{ ppm})^2 \tag{1}$$

where the $X_{CO_2}$ uncertainty from the retrieval, $\sigma_{j,retr}$, has been increased by a floor of 0.3 ppm to account for non-linearities in the uncertainty calculation; this gives $Y_j = X_j/\sigma_j$. We then detrend this weighted difference timeseries across each flight leg – subtracting either a single constant value or a linear trend $\bar{Y}(x)$, where $x$ is the along-track distance. The autocovariance function of this difference timeseries was then computed as

$$c_h = \frac{1}{N_h} \sum_{x=1}^{N_h} (Y_x - \bar{Y})(Y_{x+h} - \bar{Y}) \tag{2}$$

where $N_h$ is the number of data values falling into each distance lag bin $h$ (using an 8 km resolution) across all six flight legs. Finally, the autocorrelation coefficient values were computed as $r_h = c_h/c_o$ to give the autocorrelation spectrum shown in Fig. 1. A variety of sensitivity tests were performed, as well: 1) the correlations were computed without first dividing by the OCO-2 measurement uncertainties; 2) the data used in the analysis were restricted to pairs for which the MFLL and OCO-2 sampling times were within some threshold (3000, 1500, 1000 seconds); 3) whole flight legs of data were left out of the computation in succession; and 4) the pre-conditioning of the timeseries was switched between subtracting a constant value versus a linear trend. The only test that significantly changed the character of the spectrum significantly was the last one.

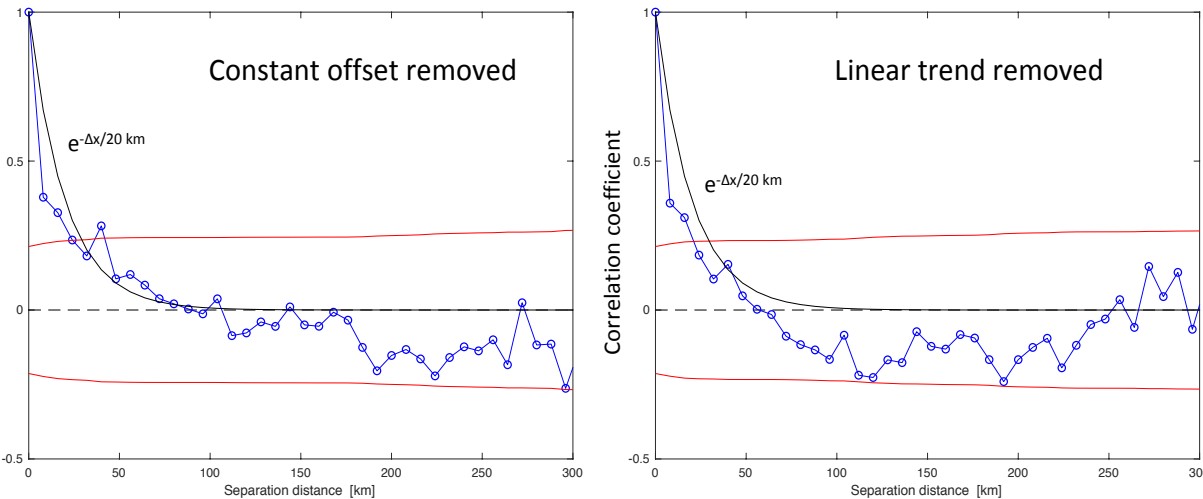

**Figure 1.** The autocorrelation of the MFLL–OCO-2 difference computed across six ACT-America flight legs as a function of the separation distance $\Delta x$ [km] along the OCO-2 ground track (blue), plus its 1-sigma significance bounds (red). Plots are given for two different preconditioning methods (subtracting off a constant value or a linear trend from each flight leg). An exponentially-decaying curve with a correlation length of 20 km is also plotted on both panels.

## 2.3 Correlation length scale results

As shown in Fig. 1, the autocorrelation of the MFLL–OCO-2 differences falls off quickly, with no significant correlations
at scales of more than about 20 km along-track. Even those correlations at length scales shorter than 20 km are only weakly
significant (at about the $1.5\,\sigma$ level); this analysis is pushing the boundaries of what this small set of comparison data can tell us.
Still, for the case in which only a constant offset between the MFLL and OCO-2 timeseries is subtracted off, an exponentially-
decaying curve with a correlation length of 20 km does a reasonable job of fitting the spectrum, although there is a tendency
for the actual spectrum to fall immediately to a correlation level of about +0.4 and then plateau somewhat out to a lag of 40 km.
When the difference timeseries are detrended with a sloping line, the correlations drop off more quickly, with a correlation
length of about 15 km, not surprisingly since more broad-scale information is removed by subtracting off the trend. We feel
that subtracting off a sloped trend from spans of OCO-2 data this short (<600 km) is not appropriate, since the actual OCO-2
bias correction procedure is done globally and certainly leaves uncorrected gradients at these scales that ought to be considered
in the analysis, so we believe that the longer length scale (20 km) would be more appropriate to apply as an OCO-2 error
correlation length. (However, when averaging over very short spatial scales, say 10 km or less, the data would then support a
correlation length scale of about 10 km, due to the rapid initial fall.)

Because the MFLL data have been blocked into bins of 7 to 9 km in length, this analysis cannot resolve scales finer than
that. That we obtain a length scale of two to three times this minimum scale does suggest that it is real and not an artifact of
the analysis. This 20 km length scale does provide a significant constraint on the information content of the OCO-2 data in an





average: it the scale was shorter, the uncertainty on longer averages would drop considerably, so if we consider the 20 km scale as an upper bound, we are being conservative (i.e. giving the data less weight in an inversion by applying a larger uncertainty to it). An exponential-decay model of the shape of the correlations is not perfect, but does not seem unreasonable to try when implementing the correlation structure found here.

## 3 Application: 10-second averages of the OCO-2 $X_{CO_2}$ data

We examine how the error correlation length scale derived above can help in weighting the column $CO_2$ data from the OCO-2 satellite when used in global flux inversions. We focus here on the specific case of averaging the data across 10-second measurement spans (equivalent to an along-track distance of ∼67.5 km), but in the process we will get insight into how to handle data assimilated at finer scales, even on an individual retrieval-by-retrieval basis. We treat the MFLL data as the "truth" and interpret the entire MFLL–OCO-2 difference as an error in the retrieved OCO-2 $X_{CO_2}$ values. An averaging approach that

considers these correlation lengths is derived below, along with alternatives that use simpler assumptions, to help illustrate the differences caused by using our new approach.

### 3.1 Data Averaging Approach

Most estimation methods used in global atmospheric trace gas inversion work (Bayesian synthesis inversions, Kalman filters, variational data assimilation) combine measurement information in different timespans as:

$$\mathbf{R}_\Sigma^{-1}\mathbf{x}_\Sigma \quad = \quad \mathbf{R}_1^{-1}\mathbf{x}_1 + \mathbf{R}_2^{-1}\mathbf{x}_2 \qquad (3)$$

$$\mathbf{R}_\Sigma^{-1} \quad = \quad \mathbf{R}_1^{-1} + \mathbf{R}_2^{-1} \qquad (4)$$

where $\mathbf{x}_i$ is a vector of measurements within timespan $i$ and $\mathbf{R}_i$ the measurement error covariance matrix for $\mathbf{x}_i$. If $\mathbf{R}^{-1}$ is thought of as measurement "information" (*i.e.*, the Fisher information matrix: see Rodgers, 2000), then $\mathbf{R}_\Sigma^{-1}$ represents the sum of the information in timespans 1 and 2, and $\mathbf{x}_\Sigma$ can be thought of as a weighted average of $\mathbf{x}_1$ and $\mathbf{x}_2$ that summarizes their

information. Note, however, that this approach veers wildly between extremes in its treatment of error correlations in time: measurement vectors $\mathbf{x}_1$ and $\mathbf{x}_2$ for different timespans are assumed to have errors that are uncorrelated with each other, while the elements of a measurement vector (at possibly different times inside the same timespan) are permitted to have any non-zero correlations, as long as they may be described by an error covariance matrix. This assumption of uncorrelated errors between different timespans is built into the derivations of these inverse methods explicitly, for example in the Kalman filter, in which

the dynamical errors related to propagating the measurement information from time to time are assumed to be uncorrelated with the measurement errors themselves. The more general case, in which errors between measurements at different times are considered to be correlated, may be written out and solved (e.g. see Bennett, 2002, Sect. 1.5.3), but the greater computational complexity and workload involved, coupled with a general lack of knowledge as to what the temporal correlations ought to be assumed to be, result in the more simple forms being used more generally.



This same philosophy of considering the error correlations between meaurements within a given time span, but neglecting them between time spans when assimilating them into inversions, has often been followed in assimilating OCO-2 $CO_2$ data in global flux inversions: the measurements are first averaged across a certain timespan (with any error correlations across these finer time/space scales being considered in the averaging process), then when these measurement averages are assimilated into the inversion, their errors are considered to be independent, in a manner similar to that used in the inversion methods

themselves.

The OCO-2 satellite makes three cross-scans per second, each of which spans only 10 km across-track and is divided into eight separate fields of view. Because the satellite "pirouettes" to keep the sun perpendicular to the viewing slit, this cross-scan is not always perpendicular to the ground track, but can come within about $20°$ of being parallel to it (Eldering et al., 2017). Thus, OCO-2 senses only a very thin swath, up to 10 km wide, but which may be as thin as only 2 or 3 km at near-sub-

solar latitudes. The satellite's FOV moves at ~6.75 km/sec along-track. Across a single second, then, OCO-2 takes up to 24 measurements across a quadrilateral with sides of 10 km and 6.75 km, flattened to different degrees around the orbit. Not all of these 24 FOVs produce reliable retrievals due to clouds, high aerosol optical depths, or other problems that prevent the scene from passing the quality filters (*xco2_quality_flag*=0 indicating a "good" scene).

Suppose we want to average the OCO-2 measurements across some distance along-track that will be closer to the grid

box size of the typical atmospheric transport model used in global flux inversion studies (100s of km on a side). The OCO-2 flux inversion MIP has averaged across a 10-second (67.5 km) swath, so we will use that here. If we form a vector $\mathbf{x} \equiv [X_1, X_2, \ldots, X_J]^T$ of J (out of 240) "good" retrieved $X_{CO_2}$ values to be averaged in our 10-second span, then their weighted average is calculated as:

$$\bar{X} = \frac{\mathbf{w}^T\mathbf{x}}{\mathbf{w}^T\mathbf{1}} = \frac{\sum_{j=1}^{J} w_j X_j}{\sum_{j=1}^{J} w_j} \tag{5}$$

with $\mathbf{1}$ being a vector of ones. Choosing weights $\mathbf{w} = \mathbf{R}^{-1}\mathbf{1}$ gives the scalar version of the sort of measurement information-weighted average shown in equation (3), which allows correlations between measurement errors within a given measurement span to be considered by specifying non-zero elements on the off-diagonal portion of the measurment error covariance matrix $\mathbf{R}$. Suppose we break out the error covariance $\mathbf{R}$ into a form that explicitly represents the correlations as $\mathbf{R} = \mathbf{SCS}$, where $\mathbf{C}$ is the correlation matrix, with ones on the main diagonal and the correlation coefficients between the elements of measurement

error $d\mathbf{x}$ on the off-diagonals, and where $\mathbf{S} = \text{diag}[\sigma_1, \sigma_2, \ldots, \sigma_J]$. The general form for the average then becomes:

$$\bar{X} = \frac{\mathbf{1}^T\mathbf{R}^{-1}\mathbf{x}}{\mathbf{1}^T\mathbf{R}^{-1}\mathbf{1}} = \frac{(\mathbf{S}^{-1}\mathbf{1})^T\mathbf{C}^{-1}\mathbf{S}^{-1}\mathbf{x}}{(\mathbf{S}^{-1}\mathbf{1})^T\mathbf{C}^{-1}(\mathbf{S}^{-1}\mathbf{1})} = \frac{\mathbf{s}^T\mathbf{C}^{-1}\mathbf{S}^{-1}\mathbf{x}}{\mathbf{s}^T\mathbf{C}^{-1}\mathbf{s}} \tag{6}$$

where $\mathbf{s} = [\sigma_1^{-1}, \sigma_2^{-1}, \ldots, \sigma_J^{-1}]^T$ and with the uncertainty in $\bar{X}$ found from

$$\sigma_{\bar{X}}^2 = \text{E}((d\bar{X})^2) = \frac{\mathbf{s}^T\mathbf{C}^{-1}\mathbf{S}^{-1}}{\mathbf{s}^T\mathbf{C}^{-1}\mathbf{s}}\text{E}[d\mathbf{x}d\mathbf{x}^T][\frac{\mathbf{s}^T\mathbf{C}^{-1}\mathbf{S}^{-1}}{\mathbf{s}^T\mathbf{C}^{-1}\mathbf{s}}]^T = \frac{1}{(\mathbf{s}^T\mathbf{C}^{-1}\mathbf{s})^2}(\mathbf{s}^T\mathbf{C}^{-1}\mathbf{S}^{-1})(\mathbf{SCS})(\mathbf{S}^{-1}\mathbf{C}^{-1}\mathbf{s}) = \frac{1}{\mathbf{s}^T\mathbf{C}^{-1}\mathbf{s}} \tag{7}$$

Whether we want to consider correlations *between* these averages $\bar{X}$ inside a given orbit is a separate matter. We might be

forgiven from considering them to be independent in light of the choices made in the estimation methods themselves, especially





if the along-track averaging length were to be long compared to the dominant error correlation length scale. But more on that later (in Sect. 3.4).

In the following subsections, we consider three different averages defined by (6), each using a different form for the correlation matrix $\mathbf{C}$.

### 3.1.1 Averaging assuming uncorrelated errors

If the data values going into the average are assumed to have independent errors, setting the correlations to zero ($\mathbf{C} = \mathbf{I}$) gives:

$$\bar{X}_{indp} = \frac{\sum \sigma_j^{-2} X_j}{\sum \sigma_j^{-2}} \tag{8}$$

$$\sigma_{indp}^{-2} = \sum \sigma_j^{-2} \tag{9}$$

The summations here and hereafter are taken over $j = 1, \ldots, J$, unless otherwize indicated, to simplify the notation. If $\sigma_j = \sigma_o$ for all $j$, this gives the well-known result that $\sigma_{indp} = \sigma_o / \sqrt{J}$.

One might use the straight information-weighted average given by (8) even in cases in which the errors are known to be correlated, but when no good model for those correlations is available. In such a case, one could calculate an average uncertainty on the mean as follows:

$$\sigma_{avg}^{-2} = \frac{1}{J} \sum \sigma_j^{-2} \rightarrow \sigma_{avg} = \sqrt{\frac{J}{\sum \sigma_j^{-2}}} \tag{10}$$

This gives an *average* uncertainty on the mean that is similar in magnitude to the uncertainties of the averaged values, rather than an uncertainty that decreases to reflect the sum of the incoming information. This former approach was the one used in the first attempt to compute 10-second averages from the OCO-2 $X_{CO_2}$ data (using the version 7 release) – see Crowell et al. (2019) for details.

### 3.1.2 Averaging assuming constant correlations not depending on distance

Suppose that the error on each retrieval inside the averaging span is correlated with the error in every other retrieval inside the span with the same correlation coefficient, $c$, such that we have the $J$x$J$ correlation matrix

$$\mathbf{C} = \begin{bmatrix} 1 & c & \ldots & c \\ c & 1 & \ldots & c \\ \vdots & \vdots & \ddots & \vdots \\ c & c & \ldots & 1 \end{bmatrix} \tag{11}$$

If we define $\mathbf{H}$ to be a $J$x$J$ matrix with ones in every element, then $\mathbf{C} = (1-c)\mathbf{I} + c\mathbf{H}$, and (noting that $\mathbf{H}^2 = J\mathbf{H}$) we get

$$\mathbf{C}^{-1} = \frac{1}{1-c}(\mathbf{I} - \frac{c}{Jc + (1-c)}\mathbf{H}) \tag{12}$$





Solving (6) and (7) with this for $\mathbf{C}^{-1}$ gives

$$\bar{X}_{flatc} = \frac{\sum x_j/\sigma_j^2 - \frac{c}{Jc-(1-c)}(\sum \sigma_j^{-1})(\sum x_j/\sigma_j)}{\sum \sigma_j^{-2} - \frac{c}{Jc+(1-c)}(\sum \sigma_j^{-1})^2} \tag{13}$$

$$\sigma_{flatc}^{-2} = \frac{1}{1-c}[\sum \sigma_j^{-2} - \frac{c}{Jc+(1-c)}(\sum \sigma_j^{-1})^2] \tag{14}$$

Based on work by Susan Kulawik (personal communication, mid-March 2019), we used the following positive OCO-2

measurement correlation values in the MIP:

$$c = \begin{cases} +0.3 & \text{over land} \\ +0.6 & \text{over water} \\ +0.6 & \text{for mixed land/water } (data\_type = 9) \end{cases} \tag{15}$$

### 3.1.3   Averaging assuming correlations that decay exponentially with distance

If an error correlation length, $L$, is known, consider a 1-D error correlation model with positive correlations of the form
$c = e^{\frac{-\Delta x}{L}}$. We will use the correlation length scale calculated for OCO-2 from the MFLL measurements for $L$. If we have $J$

points to be averaged, spaced equally in the along-track direction and separated by distance $\Delta x$, then we have

$$\mathbf{C} = \begin{bmatrix} 1 & c & c^2 & c^3 & \ldots & c^{J-1} \\ c & 1 & c & c^2 & \ldots & c^{J-2} \\ c^2 & c & 1 & c & \ldots & c^{J-3} \\ c^3 & c^2 & c & 1 & \ldots & c^{J-4} \\ \vdots & \vdots & \vdots & \vdots & \ddots & \vdots \\ c^{J-1} & c^{J-2} & c^{J-3} & c^{J-4} & \ldots & 1 \end{bmatrix} \tag{16}$$

with $\mathbf{C}^{-1}$ having the following convenient tri-diagonal form:

$$\mathbf{C}^{-1} = \frac{1}{1-c^2}\begin{bmatrix} 1 & -c & 0 & 0 & \ldots & 0 & 0 \\ -c & 1+c^2 & -c & 0 & \ldots & 0 & 0 \\ 0 & -c & 1+c^2 & -c & \ldots & 0 & 0 \\ 0 & 0 & -c & 1+c^2 & \ldots & 0 & 0 \\ \vdots & \vdots & \vdots & \vdots & \ddots & \vdots & \vdots \\ 0 & 0 & 0 & 0 & \ldots & -c & 1 \end{bmatrix} \tag{17}$$





Plugging this form for $\mathbf{C}^{-1}$ into (7) and (6) gives

$$\sigma_{clen}^{-2} = \mathbf{s}^T \mathbf{C}^{-1} \mathbf{s} = \frac{\mathbf{s}^T}{1-c^2} \begin{vmatrix} s_1 - cs_2 \\ (1+c^2)s_2 - c(s_1 + s_3) \\ (1+c^2)s_3 - c(s_2 + s_4) \\ \vdots \\ (1+c^2)s_{J-1} - c(s_{J-2} + s_J) \\ -cs_{J-1} + s_J \end{vmatrix} \tag{18}$$

$$= \frac{1}{1-c^2}\left[(s_1 - cs_2)s_1 + (s_J - cs_{J-1})s_J + \sum_{j=2}^{J-1}[(1+c^2)s_j^2 - c(s_{j-1} + s_{j+1})s_j]\right] \tag{19}$$

$$= \frac{1}{1-c^2}\left[\sum_{j=1}^{J} s_j^2 + \sum_{j=2}^{J-1} c^2 s_j^2 - c\sum_{j=1}^{J-1} s_j s_{j+1} - c\sum_{j=2}^{J} s_j s_{j-1}\right] \tag{20}$$

$$= \frac{1}{1-c^2}\left[(1-c^2)s_1^2 + \sum_{j=1}^{J-1}[c^2 s_j^2 - 2cs_j s_{j+1} + s_{j+1}^2]\right] \tag{21}$$

$$\sigma_{clen}^{-2} = \sigma_1^{-2} + \frac{1}{1-c^2}\sum_{j=1}^{J-1}\left(\frac{1}{\sigma_{j+1}} - \frac{c}{\sigma_j}\right)^2 \tag{22}$$

$$\sigma_{clen}^{-2} \bar{X}_{clen} = \mathbf{s}^T \mathbf{C}^{-1} \mathbf{S}^{-1} \mathbf{x} = \frac{\mathbf{s}^T}{1-c^2} \begin{vmatrix} s_1 x_1 - cs_2 x_2 \\ (1+c^2)s_2 x_2 - c(s_1 x_1 + s_3 x_3) \\ (1+c^2)s_3 x_3 - c(s_2 x_2 + s_4 x_4) \\ \vdots \\ (1+c^2)s_{J-1}x_{J-1} - c(s_{J-2}x_{J-2} + s_J x_J) \\ -cs_{J-1}x_{J-1} + s_J x_J \end{vmatrix} \tag{23}$$

$$= \frac{1}{1-c^2}\left[(s_1 x_1 - cs_2 x_2)s_1 + (s_J x_J - cs_{J-1}x_{J-1})s_J + \sum_{j=2}^{J-1}[(1+c^2)s_j^2 x_j - c(s_{j-1}x_{j-1} + s_{j+1}x_{j+1})s_j]\right] \tag{24}$$

$$= \frac{1}{1-c^2}\left[\sum_{j=1}^{J} s_j^2 x_j + \sum_{j=2}^{J-1} c^2 s_j^2 x_j - c\sum_{j=1}^{J-1} s_j x_j s_{j+1} - c\sum_{j=2}^{J} s_j x_j s_{j-1}\right] \tag{25}$$

$$= \frac{1}{1-c^2}\left[(1-c^2)s_1^2 x_1 + \sum_{j=1}^{J-1}[c^2 s_j^2 x_j - cs_j s_{j+1}(x_j + x_{j+1}) + s_{j+1}^2 x_{j+1}]\right] \tag{26}$$

$$\sigma_{clen}^{-2} \bar{X}_{clen} = \frac{x_1}{\sigma_1^2} + \frac{1}{1-c^2}\sum_{j=1}^{J-1}\left[\frac{c^2 x_j}{\sigma_j^2} - \frac{c(x_j + x_{j+1})}{\sigma_j \sigma_{j+1}} + \frac{x_{j+1}}{\sigma_{j+1}^2}\right] \tag{27}$$

giving

$$\bar{X}_{clen} = \frac{\frac{(1-c^2)x_1}{\sigma_1^2} + \sum_{j=1}^{J-1}\left[\frac{c^2 x_j}{\sigma_j^2} - \frac{c(x_j + x_{j+1})}{\sigma_j \sigma_{j+1}} + \frac{x_{j+1}}{\sigma_{j+1}^2}\right]}{\frac{1-c^2}{\sigma_1^2} + \sum_{j=1}^{J-1}\left(\frac{c}{\sigma_j} - \frac{1}{\sigma_{j+1}}\right)^2} \tag{28}$$



To use this 1-D error correlation model for actual OCO-2 data, which may fall as much as 5 km on either side of the center of the ground track, some averaging in the cross-track direction must be done first. Once that is done, the data could then be averaged in the along-track direction at scales of anywhere from $\Delta x = 2.25$ km (given by the 3 Hz cross-track scan frequency) all the way up to the 67.5 km distance travelled across the 10-second averaging span. To use the 1-D model with real data, for which there are data gaps, the missing data would be given $s_j$ values equal to zero in the formulas above.

The local nature of the average in (28) suggests a second use of this model: as a pre-conditioner for data to be assimilated retrieval-by-retrieval (individually, without averaging) in an inversion, with each measurement assumed to have errors independent of all the others. Currently, many inversion schemes ingest the data without averaging, retrieval by individual retrieval, sometimes inflating the uncertainties on individual data in an *ad hoc* manner to account for correlations, sometimes not. Equations (18) and (23) present a way to adjust the data beforehand, so that when it is assimilated retrieval-by-retrieval, the correlations are accounted for in a statistically-justifiable manner: pass over the data once before assimilating them, modifying each datum $x_j$ and its associated $\sigma_j$ using the two data points on either side of it, along-track, such that

$$\acute{\sigma}_j^{-2} = \sigma_j^{-1}[(1+c^2)\sigma_j^{-1} - c(\sigma_{j-1}^{-1} + \sigma_{j+1}^{-1})]/(1-c^2) \tag{29}$$

$$\acute{x}_j = \acute{\sigma}_j^2 \sigma_j^{-1}[(1+c^2)x_j/\sigma_j - c(x_{j-1}/\sigma_{j-1} + x_{j+1}/\sigma_{j+1})]/(1-c^2) \tag{30}$$

where the primes indicate the new, adjusted, values. Rather than being an approximation, this will give the same answer, when each datum is assimilated independently, that assimilating the original data, with correlations handled properly in the equations, would give. However, both methods (treating the correlations in averages or in assimilating the data individually) may suffer a serious problem, described next.

### 3.2 The problem of negative weights

In defining any weighted average of the sort assumed in equation (5), the individual weights are required to be non-negative; some could be allowed to be zero, but not all of them, so that the denominator does not go to zero. It is unfortunately the case that negative weights may occur for both the constant and exponential error correlation models that we have outlined in the previous two sections. The practical effect of negative weights is that the average value computed can fall outside of the range of the values input to the average, a feature generally not considered desirable in an average.

This very out-of-range problem was discovered when the constant correlation model from Sect. 3.1.2 was applied recently to the OCO-2 v9 $X_{CO_2}$ data by the OCO-2 flux inversion MIP team. The weights $\mathbf{R}^{-1}\mathbf{1}$ for that model are

$$\mathbf{w} = \mathbf{R}^{-1}\mathbf{1} = \frac{1}{1-c}\mathbf{S}^{-1}[\mathbf{I} - \frac{c}{Jc+(1-c)}\mathbf{H}]\mathbf{S}^{-1}\mathbf{1} \tag{31}$$

$$= \frac{1}{1-c} \begin{vmatrix} \sigma_1^{-2} \\ \sigma_2^{-2} \\ \vdots \\ \sigma_J^{-2} \end{vmatrix} - \frac{c/(1-c)}{Jc+(1-c)} \begin{vmatrix} \sigma_1^{-1} \\ \sigma_2^{-1} \\ \vdots \\ \sigma_J^{-1} \end{vmatrix} \sum_{j=1}^{J} \sigma_j^{-1} \tag{32}$$





any element of which can go negative when $\sigma_i^{-1} < \frac{1}{J-1+\frac{1}{c}} \sum_{j=1}^{J} \sigma_j^{-1}$ or

$$\sigma_i > \left( \frac{1}{J-1+\frac{1}{c}} \sum_{j=1}^{J} \sigma_j^{-1} \right)^{-1} \tag{33}$$

Thus, most uncertainties that are larger than average (an average in terms of the inverse of the uncertainty) can cause negative weights.

The exponential correlation model suffers a similar problem. Taking an interior element of equation (18) as a guide, we have

$$w_j = (\mathbf{R}^{-1}\mathbf{1})_j = (\mathbf{S}^{-1}\mathbf{C}^{-1}\mathbf{s})_j = \frac{\sigma_j^{-1}}{1-c^2}((1+c^2)\sigma_j^{-1} - c(\sigma_{j-1}^{-1} + \sigma_{j+1}^{-1})) \tag{34}$$

which goes negative when

$$\sigma_j^{-1} < \frac{c}{1+c^2}(\sigma_{j-1}^{-1} + \sigma_{j+1}^{-1}) = \frac{2}{c^{-1}+c^{+1}} \frac{(\sigma_{j-1}^{-1} + \sigma_{j+1}^{-1})}{2} \tag{35}$$

or (since $c = \mathrm{e}^{-\Delta x/L}$) when

$$\sigma_j > \frac{\cosh(\Delta x/L)}{(\sigma_{j-1}^{-1} + \sigma_{j+1}^{-1})/2} \tag{36}$$

This condition is violated only somewhat less frequently than (33): whenever $\sigma_j$ is $\cosh(\Delta x/L)$ times greater than a similar average (of the inverses) of the uncertainties of the two neighboring points. When averaging many points at finer scales ($\Delta x < L$), $1 < \cosh(\Delta x/L) < 1.54$, and up to half the points will cause negative weights.

How can we sidestep this issue for practical problems? Could one try just discarding the retrievals with higher retrieval uncertainties $\sigma_j$? In the case of the constant-correlation model, this would be impractical, as roughly half of the retrievals would have above-average $\sigma_j$ values, and if those were thrown out, half of the remainder would have to be thrown out, and so on. For the exponential correlation model, however, a sort of filtering approach might be feasible when $\cosh(\Delta x/L)$ is significantly above one (i.e., for $\Delta x$ values approaching $L$): one could throw out the retrievals with anomalously-high $\sigma_j$

values to the point that $\sigma_j$ varied smoothly enough from retrieval to retrieval to ensure that condition (36) would never be violated. This might be practicable for spans with sparser data (longer $\Delta x$ values) or when averaging together data that had already been binned together at finer scales.

As an alternative to using the negative weight criterion as a (possibly harsh) data filter, one might specify the form of the weighted average *a priori*, in a way that forces the weights to be positive, rather than letting the weights be determined

indirectly by specifying the correlation model as we have done above. For example, one could specify the form of the average to be the information-weighted mean given in (8), but then impose the correlated error assumptions of one's choice when calculating the uncertainty on that mean. Since the mean given by (8) would, in general, no longer be the optimal (minimum variance) value for that error model, one might expect that the uncertainties on the mean obtained would be higher than those given by (14) or (22).





In general, if $\mathbf{C}_w$ is the correlation model assumed in setting the weights for the average, and $\mathbf{C}_{err}$ the correlation model assumed for the actual errors $d\mathbf{x}$, then

$$\sigma^2 = E[dx dx^T] = \frac{\mathbf{s}^T \mathbf{C}_w^{-1} \mathbf{S}^{-1}}{\mathbf{s}^T \mathbf{C}_w^{-1} \mathbf{s}} E[d\mathbf{x}(d\mathbf{x})^T] \frac{\mathbf{S}^{-T} \mathbf{C}_w^{-1} \mathbf{s}}{\mathbf{s}^T \mathbf{C}_w^{-1} \mathbf{s}} = \frac{1}{(\mathbf{s}^T \mathbf{C}_w^{-1} \mathbf{s})^2} \mathbf{s}^T \mathbf{C}_w^{-1} \mathbf{S}^{-1} [\mathbf{S} \mathbf{C}_{err} \mathbf{S}^T] \mathbf{S}^{-T} \mathbf{C}_w^{-1} \mathbf{s} \tag{37}$$

$$= \frac{\mathbf{s}^T \mathbf{C}_w^{-1} \mathbf{C}_{err} \mathbf{C}_w^{-1} \mathbf{s}}{(\mathbf{s}^T \mathbf{C}_w^{-1} \mathbf{s})^2} \tag{38}$$

### 3.2.1    Constant error correlation case with sub-optimal average

When the OCO-2 flux inversion MIP group encountered this out-of-range, negative weight problem when applying equation (13) to the OCO-2 v9 $X_{CO_2}$ data, this was in fact the work-around that we fell back to: the mean was specified by (8), but the errors between individual retrievals were assumed to be correlated according to the constant error correlation model from (11) (Peiro et al., submitted). Setting $\mathbf{C}_w = \mathbf{I}$ and $\mathbf{C}_{err} = [(1-c)\mathbf{I} + c\mathbf{H}]$ allows the uncertainty on the mean to be computed as:

$$\sigma^2_{flatc2} = \frac{1}{(\mathbf{s}^T \mathbf{s})^2} \mathbf{s}^T [(1-c)\mathbf{I} + c\mathbf{H}] \mathbf{s} = \frac{1}{(\sum \sigma_j^{-2})^2}[(1-c)(\sum \sigma_j^{-2}) + c(\sum \sigma_j^{-1})^2] = \frac{1}{\sum \sigma_j^{-2}}[(1-c) + c\frac{(\sum \sigma_j^{-1})^2}{\sum \sigma_j^{-2}}] \tag{39}$$

In terms of the measurement information, this gives

$$\sigma^{-2}_{flatc2} = \frac{\sum \sigma_j^{-2}}{(1-c) + c\frac{(\sum \sigma_j^{-1})^2}{\sum \sigma_j^{-2}}} \tag{40}$$

### 3.2.2    Exponentially-decaying error correlation case with sub-optimal average

If one falls back to using (8) to calculate the means, but still uses the exponential error correlation model, then (setting $\mathbf{C}_{err} = \mathbf{C}$ from (16)) the uncertainty on the mean is computed (with $c = e^{\frac{-\Delta x}{L}}$) as:

$$
\quad \sigma^2_{clen2} = \frac{1}{(\mathbf{s}^T \mathbf{s})^2} \mathbf{s}^T
\begin{vmatrix}
1 & c & c^2 & c^3 & \dots & c^{J-1} \\
c & 1 & c & c^2 & \dots & c^{J-2} \\
c^2 & c & 1 & c & \dots & c^{J-3} \\
c^3 & c^2 & c & 1 & \dots & c^{J-4} \\
\vdots & \vdots & \vdots & \vdots & \ddots & \vdots \\
c^{J-1} & c^{J-2} & c^{J-3} & c^{J-4} & \dots & 1
\end{vmatrix}
\mathbf{s} = \frac{1}{(\sum_{j=1}^{J} \sigma_j^{-2})^2}[\sum_{j=1}^{J} \sigma_j^{-2} + 2\sum_{k=1}^{J-1} c^k \sum_{j=1}^{J-k} \sigma_j^{-1} \sigma_{j+k}^{-1}] \tag{41}
$$

$$\tag{42}$$

or in terms of measurement information

$$\sigma^{-2}_{clen2} = \frac{\sum_{j=1}^{J} \sigma_j^{-2}}{1 + 2[\sum_{k=1}^{J-1} c^k \sum_{j=1}^{J-k} \sigma_j^{-1} \sigma_{j+k}^{-1}]/\sum_{j=1}^{J} \sigma_j^{-2}} \tag{43}$$





| Correlation model | (Equation)     Result for $\sigma_j = \sigma_o$ | $\sigma_{avg}^{-2}/\sigma_o^{-2}$ |
|---|---|---|
| independent errors | (9)     $\sigma_{indp}^{-2} = J\sigma_o^{-2}$ | $J$ |
| information-averaged uncert. | (10)     $\sigma_{avg}^{-2} = \frac{1}{J}J\sigma_o^{-2}$ | 1 |
| constant correlations | (14)     $\sigma_{flatc}^{-2} = \frac{1}{1-c}[J\sigma_o^{-2} - \frac{c}{Jc+(1-c)}(J\sigma_o^{-1})^2]$ | $\frac{J}{1+c(J-1)}$     i |
| "     " , fallback weights | (40)     $\sigma_{flatc2}^{-2} = \frac{J\sigma_o^{-2}}{(1-c)+c\frac{(J\sigma_o^{-1})^2}{J\sigma_o^{-2}}}$ | $\frac{J}{1+c(J-1)}$ |
| correlation length scale | (22)     $\sigma_{clen}^{-2} = 1+(J-1)\frac{1-c}{1+c}\sigma_o^{-2}$ | $1+(J-1)\tanh(\frac{\Delta x}{2L})$ |
| "     " , fallback weights | (43)     $\sigma_{clen2}^{-2} = \frac{J\sigma_o^{-2}}{1+\frac{2}{J}\sum_{k=1}^{J-1}(J-k)c^k}$ | $\frac{J\tanh(\frac{\Delta x}{2L})}{1-\frac{1-e^{-J\Delta x/L}}{J}\text{csch}(\frac{\Delta x}{L})}$ |

**Table 2.** The analytical expressions for the uncertainty on the average given by each correlation model for a simple cases in which all measurement uncertainties have the same constant value, $\sigma_o$.

| Correlation model | (Equation)     Result for $\sigma_j^{-1} = \sigma_o^{-1}[\frac{1}{2} + \frac{j-1}{J-1}]$ | $\sigma_{avg}^{-2}/\sigma_o^{-2}$ |
|---|---|---|
| independent errors | (9)     $\sigma_{indp}^{-2} = \sum \sigma_j^{-2}$ | $\alpha$ |
| information-averaged uncert. | (10)     $\sigma_{avg}^{-2} = \frac{1}{J}\sum \sigma_j^{-2}$ | $\frac{\alpha}{J}$ |
| constant correlations | (14)     $\sigma_{flatc}^{-2} = \frac{1}{1-c}[\sum \sigma_j^{-2} - \frac{c}{Jc+(1-c)}(\sum \sigma_j^{-1})^2]$ | $\frac{1}{1-c}(\alpha - \frac{J^2c}{Jc+1-c})$ |
| "     " , fallback weights | (40)     $\sigma_{flatc2}^{-2} = \frac{\sum \sigma_j^{-2}}{(1-c)+c\frac{(\sum \sigma_j^{-1})^2}{\sum \sigma_j^{-2}}}$ | $\frac{\alpha}{1-c+\frac{J^2c}{\alpha}}$ |
| correlation length scale | (22)     $\sigma_{clen}^{-2} = \frac{1}{\sigma_1^2} + \frac{1}{1-\gamma^2}\sum_{j=1}^{J-1}[\frac{\gamma^2}{\sigma_j^2} - \frac{2\gamma}{\sigma_j\sigma_{j+1}} + \frac{1}{\sigma_{j+1}^2}]$ | $\frac{\alpha}{\tanh(\Delta x/L)} - \frac{2\gamma\beta+\gamma^2(\sigma_1^{-2}+\sigma_J^{-2})}{1-\gamma^2}$ |
| "     " , fallback weights | (43)     $\sigma_{clen2}^{-2} = \frac{(\sum \sigma_j^{-2})^2}{\sum \sigma_j^{-2}+2(\sum_{k=1}^{J-1}c^k\sum_{j=1}^{J-k}\sigma_j^{-1}\sigma_{j+k}^{-1})}$ | $\frac{\alpha}{1+\frac{2\gamma}{\alpha}[\frac{1-\gamma^{J-1}}{1-\gamma}Jf-\frac{2}{3}\frac{1}{1-\gamma^2}J^2fg+\frac{2}{3}\frac{1}{(1-\gamma)^4}h]}$ |

$$\alpha = \frac{J}{12}\frac{13J-11}{J-1}, \quad \beta = \frac{1}{J-1}[\frac{1}{4}(J-1)(J-3) + \frac{1}{2}J(J-2) + \frac{1}{6}J(2J-1)], \quad \gamma = e^{-\Delta x/L},$$
$$f = (J-3)^2 + (J+1)(10J-16)/3, \quad g = 1 - J\gamma^{J-1} + (J-1)\gamma^J,$$
$$h = (1+4\gamma+\gamma^2)g - J(J-1)\gamma^{J-1}(1+2\gamma+J(1-\gamma))(1-\gamma)^2$$

**Table 3.** The analytical expressions for the uncertainty on the average given by each correlation model for a simple case in which the inverse of the measurement uncertainty varies as $\sigma_j^{-1} = \sigma_o^{-1}[\frac{1}{2} + \frac{j-1}{J-1}]$.

## 385   3.3   Comparison of the error models for two simple cases

We look now at the uncertainty estimates that the error models above give for two simple example cases (Tables 2 & 3): one in which all $\sigma_j = \sigma_o$, a constant value, and a second in which $\sigma_j^{-1}$ varies across the sample as $\sigma_j^{-1} = \sigma_o^{-1}[\frac{1}{2} + \frac{j-1}{J-1}]$. Both cases may be solved analytically.

In Fig. 2, we plot $\sigma_{avg}^{-2}/\sigma_o^{-2}$ for each of these correlation models as a function of the data spacing $\Delta x$ for the two simple example cases. For the exponential correlation model, where $c = e^{-\Delta x/L}$, we divide the 10-second swath into equal increments $\Delta x = (67.5 \text{ km})/J$. For the other error models, which use the number of averaged values $J$ rather than $\Delta x$, we calculate $J = 67.5 \text{ km})/\Delta x$ for each point on the x-axis, an assumption that forces all the data to be equally spaced along-track for



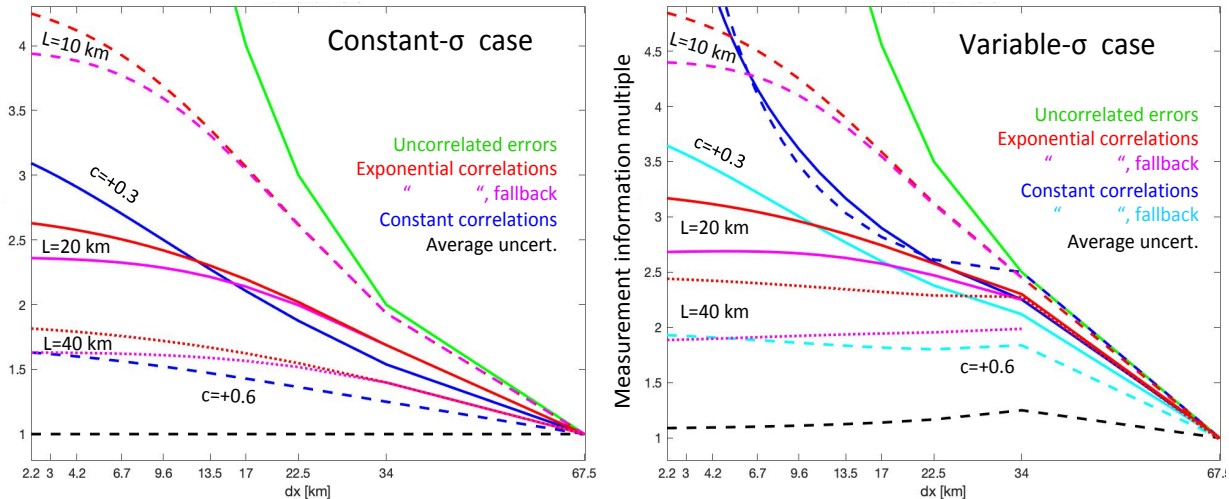

**Figure 2.** The measurement information content ($\sigma_{avg}^{-2}/\sigma_o^{-2}$) of the 10-second average produced by several different error correlation models, as a function of the linear separation of the data points, $\Delta x$, for two simple example cases: all averaged values having the same uncertainty ($\sigma_j = \sigma_o$) (left), and the values having uncertainties that vary as $\sigma_j^{-1} = \sigma_o^{-1}[\frac{1}{2} + \frac{j-1}{J-1}]$ (right). For those models based on the number of data points, $J$, we convert to $\Delta x = (67.5 \text{ km})/J$ for plotting on the x-axis. The correlation models are: independent errors (green), average uncertainty (black dashed), constant correlations (blue), and constant correlations using the fall-back uncorrelated mean (cyan), assuming for the latter two models correlations of $c$=+0.3 (solid) and $c$=+0.6 (dashed). Also given are the exponential correlation model results with the original average from (28) (red) and with the fall-back uncorrelated mean (magenta), in both cases for length scales of 10 km (dashed), 20 km (solid), and 40 km (dotted).

those models, too, in these plots. The average uncertainty (dashed black) and independent error (green) models bound the possible information range as a function of $\Delta x$, with the independent case setting the maximum. There is more variation

with $J$ for the variable uncertainty case: for both simple models, there is a tendency for the constant-correlation model to provide a more-rapidly increasing measurement information at smaller $\Delta x$ than in the exponential correlation model: this is to be expected, since the correlations in the latter when $\Delta x$<20 km are higher than the +0.3 correlation level used in the constant correlation model. In general, the constant and exponential correlation models provide similar results, other than at the smallest $\Delta x$. The total information content of the average is limited by the correlations to only about three times that of the

typical individual measurement, when assuming an error corrleation length scale of 20 km.

The results for the fallback approaches for the constant- and exponential-correlation models that we derived in Sections 3.2.1 and 3.2.2 to get around the negative weight problem (in which we specified the weighted averaged to be (8) then computed its uncertainty using the correlated errors), are also shown in Fig. 2. For the exponential correlation model, the fall-back approach (magenta lines) results in only slightly less information (or higher uncertainty) for the average, compared to the original

approach (red lines). For the constant-correlation fallback model (cyan lines), the loss of information is greater, though this is seen only in the variable-$\sigma$ case, the two models giving the same result in the constant-$\sigma$ case.



### 3.4  Calculating correlations between averaging spans

Above we have accounted for correlated errors between the retrievals going into the 10 second averages. The same 1-D error correlation model developed above can also be used to compute the correlations between adjacent 10 second average spans, if an error correlation length scale is known. In that case, the spacing $\Delta x$ between the data assumed in Sections 3.1.3 and 3.2.2 is no longer the spacing between individual retrievals but rather the spacing between the different 10 second average spans, along the ground track of the satellite.

Suppose we look at the day-lit side of a single OCO-2 orbit, which we will assume encompasses a third of the full orbit, or about 13358 km; there are about 198 10-second spans inside it. If we let $\Delta x = (13358 \ km)/J$, then we may use equations (22) and (43) again to compute the total information across the orbit. Figure 3 plots this information content for the two simplified error cases used above as a function of $\Delta x$: for our 10-second averages, we look at $\Delta x$=67.5 km, finding that the total information multiple is about 200 for the uncorrelated error case (green line), close to the $J$=198 value, as it should be. By comparing the information values for the uncorrelated error case (green line) to the values from the exponential correlation cases (red & magenta lines), we can see the impact of the correlations in reducing the total information content in the lower $\Delta x$ range. Curves for three different correlation lengths (10, 20, and 40 km) are given; curves for measurement errors for the two different formulations of the mean (from equations (22) and (43), the red and magenta curves in Fig. 2) are both plotted, but fall on top of each other in this $\Delta x$ range. By comparing the information assuming no correlations (green) versus exponential correlations (magenta), one can derive a single scalar multiple (greater than 1) of the uncertainties on the 10-second averages, that will permit the 10-second averages to be assimilated in an inversion scheme with the assumption that they are all independent, but will yield the same amount of total measurement information entering the problem as would have been obtained if the correlations had been accounted for properly using the original uncertainties. Using such an inflation factor is generally much easier than implementing the machinery for accounting for the correlations properly in the inversion code.

### 3.5  Application of the error correlation models to OCO-2 v10 $X_{CO_2}$ data

To apply the exponential error correlation models presented in Sections 3.1.3 and 3.2.2, one must account somehow for the fact that the real OCO-2 data are not one-dimensional, but fall up to 5 km on either side of the center of the OCO-2 FOV ground track. As Fig. 2 shows that there is little independent information obtained from data spaced at $\Delta x$ values much below the correlation length scale $L$, one would be justified in computing an average value at scales that fine using the independent error equation (8) and then applying a constant-correlation-based uncertainty to it using (40). To match the cross-track dimension (10 km) somewhat, we average all the OCO-2 v10 $X_{CO_2}$ data values (Baker et al., 2020) falling inside each 2-second span (spaced $\Delta x$=13.5 km apart, along-track) in this manner. We could then apply any of the error correlation models that we have discussed so far (constant or exponential, using the original or fallback forms for the weighted means) to the five 2-second average values falling inside each 10-second span to get the 10-sec averages. As shown in Fig. 1, the MFLL-OCO-2 spectrum suggests that the error correlations fall off even more rapidly at small $\Delta x$ than the 20 km correlation length that we decided was the best fit to the data across the full $\Delta x$ range would suggest: we use a 10 km correlation length in calculating these





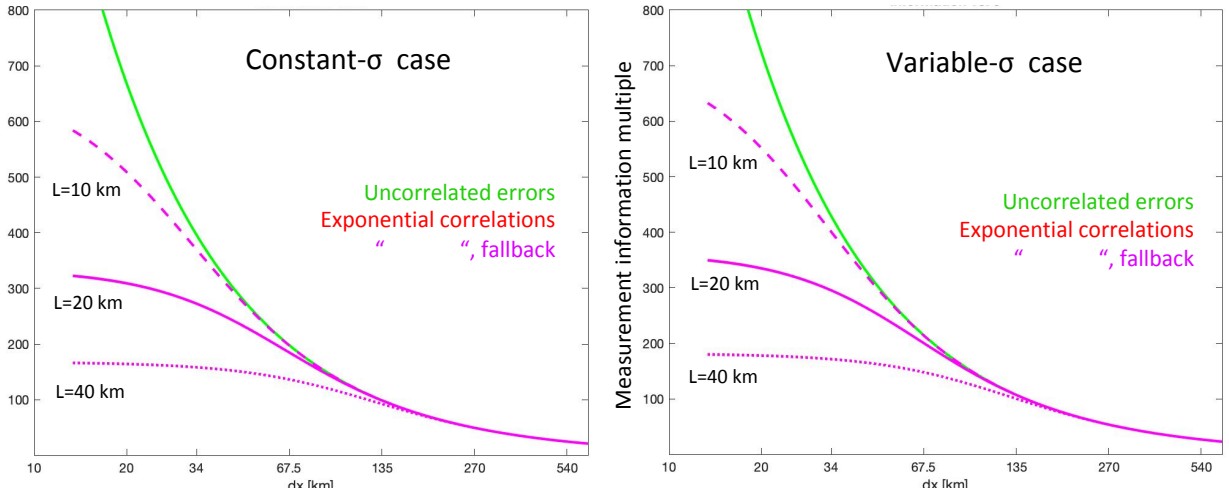

**Figure 3.** The measurement information content across 13358 km, or 1/3 of an orbit (the typical day-lit portion across which data is taken), given as $\sigma_{avg}^{-2}/\sigma_o^{-2}$, as a function of the distance $\Delta x$ spanned by each averaging interval. The number of averaging spans, $J$, is given by (13358 km)/$\Delta x$. If all averaging spans are assumed to have independent errors (green line), the information ratio equals $J$; i.e., the information is simply summed up. When the exponential correlation model is assumed (magenta lines), the total information is reduced, how much so depending on the correlation length assumed: 10 km (dashed), 20 km (solid), or 40 km (dotted). For these longer $\Delta x$ values, both assumptions for the weighting of the data going into the average for the correlation length model give the same results to the eye on this plot.

2-second averages to reflect that. An average $\Delta x$ inside the 10x13.5 km averaging box would be about 6 km (keeping in mind the stretching of the box that occurs due to the pirouetting), giving an average correlation coefficient of $e^{-6./10.}$=+0.55. Our MFLL–OCO-2 data based correlation estimates were taken over land and should apply only there: over the ocean, longer correlation lengths are generally assumed to apply. Since we don't have any MFLL data over the oceans as a guide, we will just use a correlation length double that over land there, as a guess, giving $e^{-6./20.}$=+0.74.

Once the 2-second averages are computed, they then may reasonably be averaged up to 10-second values using our 1-D error model. This is done here for both the constant-correlation and exponential correlation models, using the same OCO-2 v10 $X_{CO_2}$ data (Baker et al., 2020). Because of the difficulty satisfying the condition that all weights remain positive in the full constant-correlation case presented in Sect. 3.1.2, we will not calculate results for that model but rather only for the fallback model presented in Sect. 3.2.1. We will compute this case using both the 2-step process (averaging first across 2-second spans,

then averaging those across 10-second spans) used for the exponential correlation models, as well as using a 1-step process of averaging all retrievals inside the 10-second span in a single step, without applying the shorter correlation length scales inside the 2-second boxes. For the exponential correlation models, we will calculate results for both the original model (from Sect. 3.1.3) with the data screened to satisfy the positive weight criterion, as well as for the fallback model (from Sect. 3.2.2).

    Since we consider the exponential correlation model as an improvement to the constant-correlation model (with the fall-

back weights) given in Sect. 3.2.1, we are interested in how much this new model shifts the average from the old one. We also

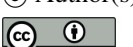

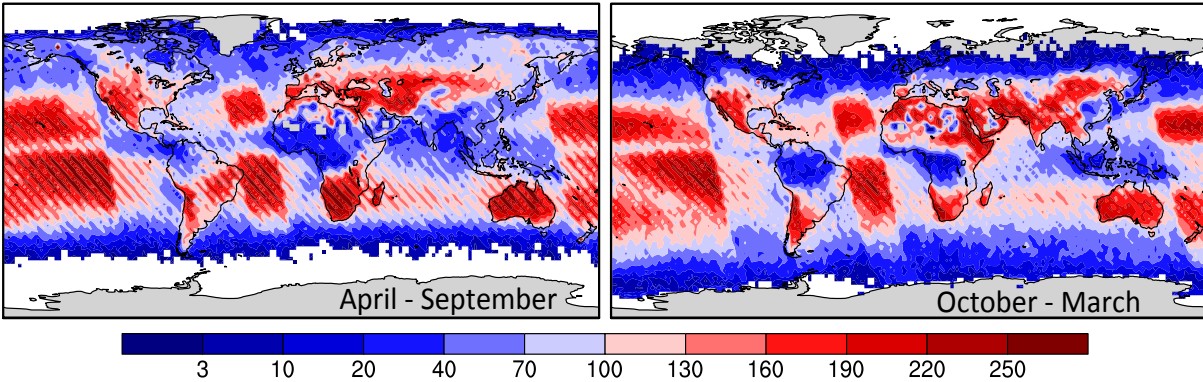

**Figure 4.** The number of 10-second average spans with good data falling within each $2°x2°$ bin across Sept 2014 - Oct 2020, for April-Sept and Oct-March.

compute the difference in the uncertainties for the four cases. We double the correlation length over the oceans from 20 km to 40 km for the calculation of the 10-second averages, as well.

All 2-second spans without any data have their $\sigma_j^{-1}$ values set to zero. Those 2-second $\sigma_j$ values that produce negative weights according to (36) have their $\sigma_j^{-1}$ values set to zero as well, to ensure that the average value stays within the range

of the input values. The 2-second average values thrown out by this approach will tend to be those with higher uncertainties sandwiched in between adjacent spans with lower uncertainties; those next to a span with no data are less likely to be discarded. Over the ocean, where the correlation length is doubled, with $\cosh(13.5/40)=1.057$, it could be expected that more of the 2-second averages will be discarded than over land, where $\cosh(13.5/20)=1.24$. That was, in fact, the case. However, uncertainties for adjacent 2-second spans do not seem to vary too much: only about 2.5% of the ocean scenes needed to be discarded and

less than 1% of them over land, for the 2-second averaging spans. The 2-second span length was chosen over the 1-second to avoid having to throw out more data that this: for the 1-second span, ~10% and ~7% of the ocean and land data would have had to be discarded, respectively.

Figure 4 shows the number of OCO-2 10-second averages per $2°x2°$ bin across September 2014 - October 2020, broken into two halves of the year: Apr-Sep and Oct-Mar, while Fig. 5 gives the 10-second average $X_{CO_2}$ values for the same spans for

the constant-correlation model from Sect. 3.2.1 acting upon the 2-second means (i.e. using the 2-step approach from above). Figure 6 then gives the difference in 10-sec averaged $X_{CO_2}$ from this 2-step model for both the 1-step constant-correlation model and the exponential correlation model. It is interesting that the exponential model, which uses completely different assumptions about the correlations as a function of $\Delta x$, gives almost identical $X_{CO_2}$ averages as the 2-step constant-correlation model. The difference to the constant-correlation averages caused by using the intermediate step of averaging the data across a

2-second span is a larger effect. (Recall that we had to use a 2-step process for the exponential correlation model, to be able to apply the 1-D error model to it in the first place. So we compute results for a similar 2-step constant correlation model as well, to allow a more accurate comparison of the two approaches.) The close agreement between the exponential model average and





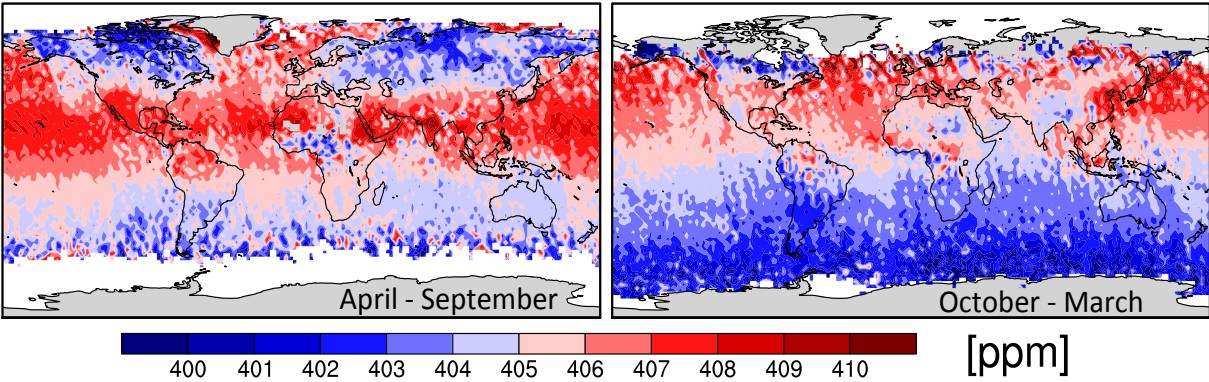

**Figure 5.** The 10-sec average $X_{CO_2}$ value for Sept 2014 - Oct 2020, binned into $2°x2°$ boxes and for April-Sept and Oct-March, given by the 2-step averaging approach (an information-weighted average of five 2-sec averages).

the 2-step constant model average is somewhat deceptive, though: when the same comparison (not shown) was done using 1-second averages for both, there was a systematic difference between the averages over the land and ocean of over 0.1 ppm,
of opposite sign. This shift may be related to the greater number of points being thrown out for causing negative weights in the 1-second averaging case.

Figure 7 presents the uncertainty of the 2-second $X_{CO_2}$ averages going into the 2-step constant-correlation averages (calculated from the uncertainties for the individual retrievals using (39)), while Fig. 8 gives the ratio of total measurement information given by four different 10-second correlation models with respect to these 2-sec uncertainty values. These information
multiples are the same as those plotted on the y-axis of Fig. 2 for the two simple error cases, and are meant to show the number of independent pieces of information allowed by the correlations across the 10-second span (a value between 1 and 5). Both of the models that addressed the negative weight issue by falling back to the independent-error average of (8), shown in the right column of Fig. 8, are similar, and yield slightly more information (or lower uncertainties on the 10-sec averages) than does the 1-step constant correlation model (upper left). The 2-step constant correlation model gives somewhat more information over
the oceans than the 1-step model, despite imposing higher correlations during the 2-second averaging: greater weight given to 2-second spans with sparser data may explain the difference. The exponential correlation model (lower left) gives significantly higher measurement information content (lower uncertainty) than the other three correlation models, especially over the oceans. The large difference over the oceans is in agreement with what is seen in Fig. 2 for the simple model with variable $\sigma_j$ (right panel), where the dotted orange line (exponential model for $L$=40 km) at $\Delta x$=13.5 km is about 25% higher than the
dashed cyan line (showing the constant-correlation result for $c$=+0.6). In contrast, the values over land are similar between the models in that figure: compare the solid orange and magenta lines (for the original and fallback exponential models, assuming $L$=20 km) to the solid cyan ($c$=+0.3 constant correlation) curve. The information loss in going from a correlation length of 20 km over land to 40 km over the ocean in the exponential model is not as large as the loss of information in going from a

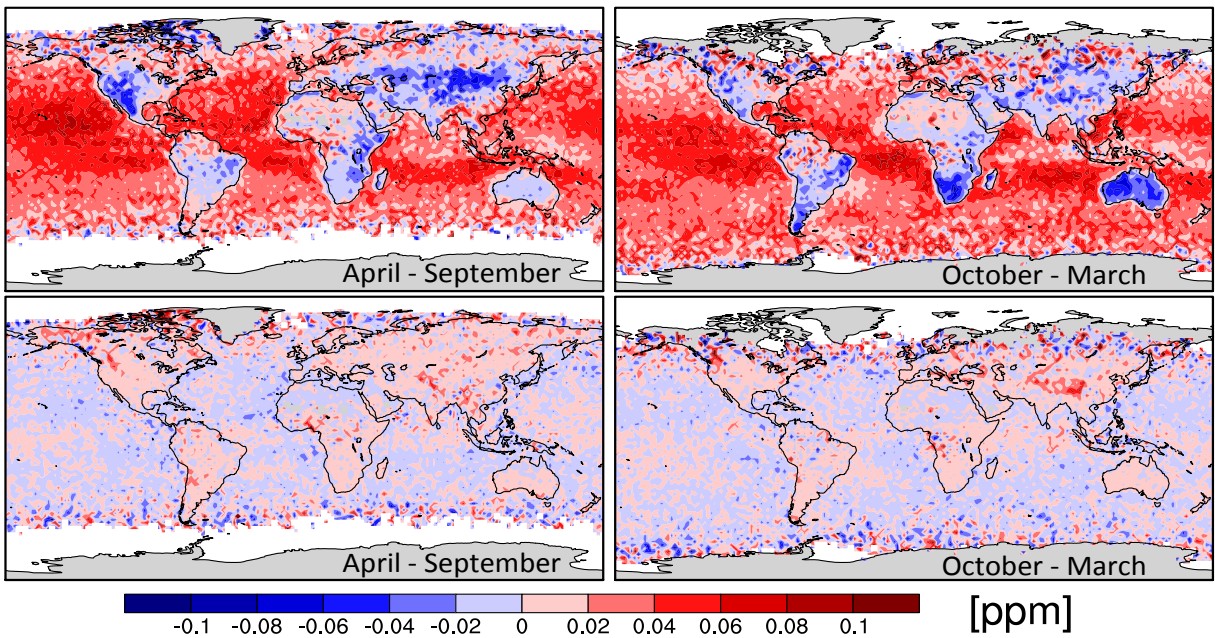

**Figure 6.** The difference or shift [ppm] in $X_{CO_2}$ from the 2-step average shown in Fig. 5 given by the 1-step information average (given by summing all the good scenes within each 10-sec span without first computing 1-sec averages)(top) and by the exponential correlation model (bottom), for April-Sept and Oct-March.

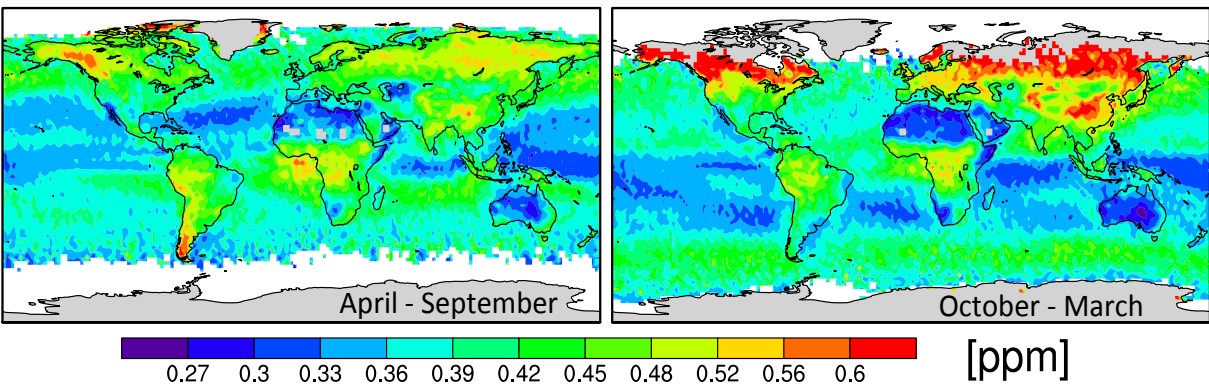

**Figure 7.** The uncertainty [ppm] in the 2-second average $X_{CO_2}$ values given by the constant correlation model using the information-based weights (equation (39)) and the half correlation lengths (see text) for April-Sept and Oct-March.

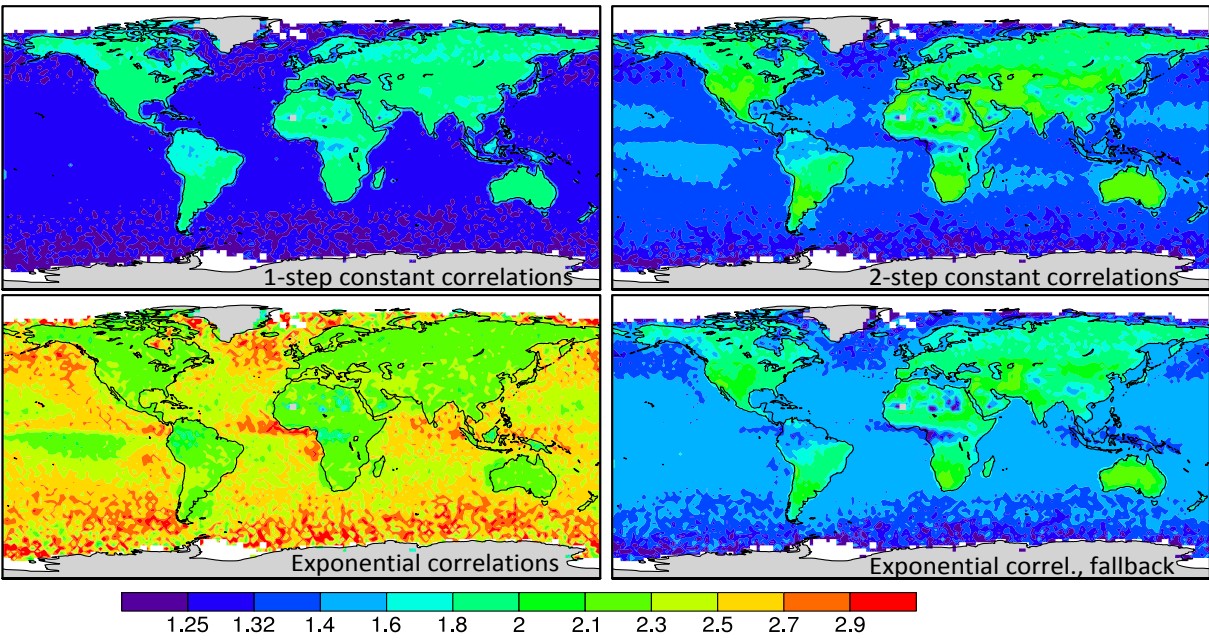

**Figure 8.** The ratio of the measurement information, defined as $\sigma^{-2}$, for the $X_{CO_2}$ 10-second average given by four error correlation models over the measurement information of the 2-second average values shown in Fig. 7, computed across Sep 2014 - Oct 2020 (annual, not half yearly). The four correlation models are the 1-step and 2-step constant correlation models (upper left, upper right) and the exponential correlation model using (8) and (28) to calcuate the average (bottom left, bottom right).

correlation of +0.3 to +0.6 in the constant correlation model, at least at $\Delta x$=13.5 km: that explains the different character of

the results over the oceans versus land.

The exponential correlation model seems to be quite sensitive to data dropout: if every other 2-second average along-track is thrown out, the total information returned in the average goes up, as can be seen from equation (21). This is counter-intuitive and is a feature of this model that deserves more investigation.

## 4   Summary and Conclusions

It has long been recognized in the atmospheric modeling community that "measurement errors" in flux inversions must include not just instrumentation errors, but also errors incurred when representing the measurements in the models, especially the coarse-resolution transport models used in global inversions. In the case of *in situ* measurements, the former might be on the order of 0.1 or 0.2 ppm, while the latter could range from as low as 0.2 ppm in the remote Southern Hemisphere to multiple ppm for continental sites farther north, especially those feeling the effects of nearby forests or cities. Correlations between

measurements located near to each other in space or time should similarly be due to modeling errors in these "model-data mismatch" (MDM) errors. In inversions using just *in situ* data, the MDM errors would be increased to deweight multiple





sites located close to each other (e.g., Bermuda East and West), or multiple data streams from different measurement groups or different types of sensors at a single site (Mauna Loa, Cape Grim, South Pole). To account for diurnal modeling errors, continuous measurements would be deweighted versus daily measurements, and both against weekly flask measurements.

Aircraft profiles spanning multiple vertical levels might be deweighted to account for vertical mixing errors.

For satellites such as GOSAT, which take discrete column-averaged measurements spaced generally over 100 km apart from each other around the orbit, the need for modeling correlations between measurements was less immediate: a modeling error could be added to the retrieval error in quadrature in a plausible error treatment. For a satellite like OCO-2, however, with up to a thousand measurements taken in a thin swath across a 300 km span, there are sure to be correlations in the retrieval errors

on those scales (due to the parameters assumed in the retrievals, and the modeling errors on them, varying on those scales), not to mention atmospheric modeling errors, as well.

For the OCO-2 version 7 $X_{CO_2}$ data, the OCO-2 flux inversion MIP team used a two-step averaging approach: the data were first averaged across a 1-second span using an information-average (eq. (8)), and then an uncertainty was placed upon this value that was a combination of the average uncertainty given by (10), derived from the retrieval uncertainties, and the standard

deviation of the retrieved $X_{CO_2}$ values going into the average. Those 1-second spans with good data were then averaged using (8) with an uncertainty placed upon the average that assumed that each 1-second span had errors independent of all the others, according to equation (9). Finally, a modeling error was added in quadrature to the MDM error so calculated: this modeling error ended up being so large compared to the measurement-derived error that, in practice, the details of the measurement-derived part did not matter much. See Crowell et al. (2019) for details.

For version 9 of the OCO-2 retrievals, the OCO-2 flux inversion MIP team attempted to do a better job modeling correlations between the $X_{CO_2}$ values. Based on how quickly the difference between the OCO-2 $X_{CO_2}$ values and nearby TCCON $X_{CO_2}$ values was reduced as adjacent OCO-2 retrievals were averaged at progressively longer scales along-track, Kulawik estimated that the OCO-2 retrievals over land were correlated with a flat +0.3 coefficient, and those over the ocean at +0.6: no dependence of these values on separation distance was calculated. The MIP team derived the constant-correlation model

outlined in Sect. 3.1.2 and attempted to apply it to the v9 data, but discovered that it yielded an average value that often fell outside the range of the input values, and determined that this was due to the weights on individual terms in the average being calculated to be negative by the model. As a workaround, the fallback model presented in Sect. 3.2.1 was used instead: the average was calculated using the old information average of (8), with a newly-calculated uncertainty given by (39), using the constant correlation coefficients. The average was performed in a single step across the 10-sec span, without first computing

averages at shorter spans. (This was because Kulawik et al. (2019) had found that giving the sparser OCO-2 data more weight via the previous 2-step approach resulted in a poorer fit to the TCCON data: the sparser data, while providing better spatial coverage, also are apparently more susceptible to errors due to nearby clouds or other scatterers.)

The MFLL–OCO-2 differences that we have analyzed here suggest that the assumption that the OCO-2 error correlations are constant across scales all the way up to the 67.5 km 10-second averaging length is not a good one: if the MFLL–OCO-2





differences are, in fact, a good proxy for errors on the full $X_{CO_2}$ column[1], then Fig. 1 shows that their correlations drop off rapidly, with a decorrelation length of about 20 km fitting the data well across most of the measured spectrum. These data also suggest that much of the correlation in the differences falls off even more rapidly, to a coefficient of below +0.4 at the finest resolvable scale of 8 km, corresponding to a correlation length of more like 10 km at these finest scales. These two scales might be due to separate error sources at the two different $\Delta x$ scales: for example, more-quickly changing errors due

to surface-related parameters in the retrieval (albedo, pressure over topography) versus more-slowly changing errors due to atmosphere-related parameters such as water vapor, temperature, or aerosols.

With these correlation length scales in hand, we derived here a one-dimensional error model with exponentially-decaying correlations as a function of the along-track distance (Sect. 3.1.3). We discovered that this model suffers from the same negative weight problem that our constant-correlation model did, though to a lesser extent, and derived a similar fallback model as was

used with the constant-correlation approach as one way to get around this problem (Sect. 3.2.2). We applied all these correlation models to two simple cases that could be solved analytically, to get a better understanding of how the models behaved with different averaging assumptions (different separations $\Delta x$ of the incoming average values).

Finally, we applied both our new exponential model and our old constant-correlation model to the recently-released OCO-2 version 10 $X_{CO_2}$ data. For the exponential model, we first averaged the data across a 2-second span, to force the cross-track

retrievals onto the 1-D satellite FOV ground track. By choosing this pre-averaging span to be 2 seconds long, we were able to reduce the number of spans that had to be thrown out for violating the positive weight criterion to just a couple percent, allowing the original exponential model (from Sect. 3.1.3) to be applied for a subset of the data similar to that used in the other error models. The new exponential model caused generally negligible shifts in the average $X_{CO_2}$ values as compared to the constant-correlation model at seasonal to annual scales. This is good news for our previous results and suggests that the constant

correlation coefficients that we used in the OCO-2 v9 MIP studies (Peiro et al., submitted) do a good job approximating the correlations across the 10-second spans, on average, even though those appear to vary as a function of the distance. Shifts in $X_{CO_2}$ for individual retrievals are found at the level of a few tenths of ppm, and systematic shifts in seasonal and annual $X_{CO_2}$ between land and ocean of from 0.1 to 0.2 ppm were found when pre-averaging across 1- rather than 2-second spans.

Since the MFLL data were only taken over land, they cannot say what the OCO-2 $X_{CO_2}$ correlation length scale might

be over the oceans. As a guess, we have simply doubled the land values here to get ocean values. For that assumption, the exponential correlation model gives uncertainties on the 10-second averages that are significantly lower than those given by the constant-correlation model using the +0.6 coefficient value. Possibly this indicates that we should use longer correlation lengths over the ocean than simply doubling the land values. But from the differences seen between the constant and exponential correlation models at coarser scales in the right panel of Fig. 2, this may indicate that the new exponential correlation model is

providing better uncertainty estimates (ones which, when used in the inversions, will give more weight to the OCO-2 data with

---

[1]There are good reasons why the MFLL–OCO-2 difference might not be a good proxy for errors in OCO-2 $X_{CO_2}$ retrievals: much of the difference might be due to errors in the MFLL retrieval, or to problems matching the two measurement types in time and space, or to errors in accounting for the different vertical averaging kernels; also, only the lower 2/3 of the OCO-2 column may be compared to the MFLL data, given the cruise height of the aircraft.





respect to the prior information). Lidar underflight data similar to the MFLL data used here, but collected over the ocean, are much needed to help extend the utility of this approach globally.

The 1-D error model with exponentially-decaying correlations that we have derived here also has immediate application in two other areas: 1) in the calculation of correlations between adjacent 10-second averaging spans, and 2) in the treatment
of correlated satellite data in those inversion schemes that choose not to average the individual retrievals before assimilating them, but choose to handle the correlations inside the scheme itself. Regarding point 1), by comparing the uncertinties given by the exponential correlation model (with $\Delta x$ equal to the distance between the 10-second span, or whatever longer averaging span is being considered) to those given by assuming that the error in each 10-second span is independent from all others (i.e. by comparing the magenta and green curves from Fig. 3), an overall inflation factor may be computed which, when
the uncertainties on individual 10-sec average values are multiplied by this factor, will allow them to be assimilated in the inversion assuming independent errors, but will still give the proper weight *vis-a-vis* the prior information in the problem that would have been obtained had the correlations between 10-second averages been modeled explicitly in the inversion. Similarly, regarding point 2), equations (30) and (29) provide a way to adjust the pre-averaged measurements (the 2-second averages in our presentation here) and their uncertainties in a manner such that when each of these measurements is assimilated in the inversion
*assuming independent errors*, the same answer is obtained as if the correlations between the errors in each measurement were explicitly modeled in the inversion itself (*e.g.*, by using off-diagonal terms in $\mathbf{R}$). We make this argument using the 2-second averages instead of the original retrievals themselves because the original retrievals are not strictly laid out in a one-dimensional string, but have a distribution extending up to 10 km in the cross-track direction that must be dealt with before the 1-D model can be applied.

The one-dimensional model that we have examined here can provide some insight that might be useful for satellites whose data extends across wider swaths, like the proposed CarbonSat mission, or to missions that scan full continents in a truly two-dimensional manner (GeoCarb). As Fig. 2 shows, the correlation length scale sets a limit on the total number of independent pieces of information contained across a certain span: as a very rough approximation, the number of independent pieces of information is about equal to the total averaging span (e.g. 67.5 km) divided by the correlation length scale. In going to the
more general 2-D problem, this suggests that the total information across a given area is proportional to the number of squares, of the correlation length wide, that will fit into the area. The uncertainties of individual shots within the area may then be scaled such that their total information, when they are assimilated independently, equals that given from the correlation length analysis.

*Code and data availability.* The data and code used in this analysis may be downloaded from the CERN-based Zendolo archive at https://doi.
org/10.5281/zenodo.4399884. This includes co-located MFLL and OCO-2 column $CO_2$ measurements for six underflights and the MATLAB script used to calculate the autocorrelation spectrum from their differences, as well as OCO-2 version 10 $X_{CO_2}$ data and the FORTRAN programs used to calculate the 2- and 10-second averages from them.



*Author contributions.* JD and BL have designed, implemented, developed, and tested the MFLL instrument, as well as helped with its proper usage on the ACT flight campaigns. JC helped with the usage of MFLL on the ACT campaigns, and evaluated its performance. KD designed

and oversaw the ACT campaigns. EB obtained MFLL and OCO-2 data, co-located it in space and time, accounted for the differences in vertical averaging kernel between them, and produced files with them side by side. DFB computed the correlation spectrum of the MFLL–OCO-2 differences and developed/applied the error correlation models in the remainder of the document.

*Competing interests.* The authors declare that they have no conflict of interest.

*Acknowledgements.* DFB acknowledges support through NASA grant NNX15AJ07G, "Atmospheric Carbon and Transport study – America

(ACT-America)". DFB would like to thank the OCO-2 flux inversion MIP team for many discussions regarding the use of the constant error correlation models presented in Sections 3.1.2 and 3.2.1. In particular, he would like to acknowledge the help of Dr. Sourish Basu, who co-derived the constant error model presented in Sect. 3.1.2, discovered that using it led to out-of-range average values when it was first applied to the OCO-2 v9 $X_{CO_2}$ data, and pointed out that this was due to the weights in the average going negative; Dr. Susan Kulawik, for her work in deriving the constant error correlation values in equation (15), and for encouraging the publication of the correlation length scale

work presented in Sect. 2; and Drs. Jonathan Hobbs, Hai Nguyen, and Vineet Yadav, for pointing out that one is always free to define the weights in an average as one wishes, then to apply an error model when computing the uncertainty in that defined average – this led to the error models presented in Sections 3.2.1 and 3.2.2, the first of which was adopted by the OCO-2 flux inversion MIP team in the work using the OCO-2 v9 data. The authors would also like to thank the ACT-America, MFLL, and OCO-2 teams for collecting and processing the data used here and making it available to other researchers.



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
