# Peer review of "A new exponentially-decaying error correlation model for assimilating OCO-2 column-average $CO_2$ data, using a length scale computed from airborne lidar measurements"

_Geoscientific Model Development, 2020_

## Author Comment (AC1)

We'd like to thank the reviewers for their helpful comments. Those addressing the issue of negative weights in our average and how those relate to the error covariance matrix were particularly helpful to our understanding. We have updated our text to reflect this updated viewpoint.

Responses to comments from Reviewer #1

> The authors study the correlations between errors in
> $X_{CO2}$ satellite retrievals, based on reference lidar
> measurements, and discuss various ways to account for
> them in atmospheric inversions. The paper looks a bit
> like the clean minutes of a brainstorming meeting:
> every sentence is well written but the logical flow is
> curvy and difficult to follow. The authors have not
> done enough to make their thoughts accessible and to
> take the text beyond elaborate speculation, perhaps
> simply because their thinking is not yet ripe for
> publication. Maybe it doesn't matter, as the paper
> will be cited anyway given the role of this activity
> for the OCO-2 team, but for the few who will bother
> to read it, it may be a daunting task, perhaps in the end wasted.

We hope our replies to the detailed comments below will address this reviewer's issues.

> I am listing a number of comments here to help clarify the presentation.

>   Footnote 1, p. 25: the disclaimer here is a bit hidden, but
> it is actually essential. Basically, if the "good reasons" listed
> here are correct, all results of the paper can be ignored. This
> observation could be fatal for the patient reader who painfully
> reaches this page… In the end, nothing is given to convince the
> reader that the MFLL-OCO-2 differences do indeed represent OCO-2
> errors, that the two scales of correlation lengths found (10 and
> 20 km) should be used at all in OCO-2 error models. It's embarrassing.

No need for embarrassment here. We have attempted to calculate a correlation length scale using the limited data that is available for that purpose. When more data become available to test our conclusions, that should certainly be done -- if the reviewer can suggest some additional data that we could use to refine these estimates, we would be happy to work with it. In the meantime, the derivations presented here using the newly-calculated correlation length scale remain valid regardless of what precise value is used for that quantity. We do hope that the reviewer does not mean to suggest that no one can publish using new data -- little progress would be made in such a world.

> There is good and interesting math here, but the authors belittle
> it by arbitrarily rejecting certain math results: why should the
> negative weights not be physical (l. 20) or considered undesirable
> (l.331)? They simply follow from the authors' correlation model:
> if the authors are not satisfied with this consequence, they should
> change the model rather than fooling the math.

Based on this comment as well as one of Reviewer #2's below, we have looked into this issue more and we agree that there is nothing inherent in the structure of the error covariance matrix that precludes the individual weights in our weighted average from assuming negative values – for the error correlation models that we use, these negative values are to be expected. The main constraint provided by a positive definite covariance matrix is that the *sum* of all the weights must not be negative or zero – if that criterion is met, a weighted average may always be calculated (since this implies that the inverse of the covariance matrix is also positive definite, ensuring that the denominator in (6), $\mathbf{1}^T\mathbf{R}^{-1}\mathbf{1}$, is always positive).

In terms of computing our weighted average, the implication of this is that there is nothing in our correlation models to prevent the mean value from falling outside the range of the values going into the average. If we wish to enforce that more stringent criterion, then we may require that each individual weight be non-negative, as we have done in the paper.

While there apparently is no fundamental requirement that the individual weights in a weighted average are non-negative, one often sees this given as a basic requirement. Wikipedia, while not an unimpeachable academic reference, reflects broader practice when they give on their "weighted arithmetic mean" page the mathematical definition of such a mean as follows:
"Formally, the weighted mean of a non-empty finite multiset of data { $x_1$, $x_2$, … , $x_n$ } , with corresponding *non-negative* weights { $w_1$, $w_2$, … , $w_n$} is
<x> = sum{$w_i x_i$} / sum{$w_i$}  ...
Therefore, data elements with a high weight contribute more to the weighted mean than do elements with a low weight. *The weights cannot be negative.* Some may be zero, but not all of them (since division by zero is not allowed)." [emphasis ours]

The practical effect of any individual weight being allowed to be negative is that the mean value computed may fall outside the range of the data going into the average – a result that is contrary to the very idea of an average. We are certainly justified in imposing the requirement that all individual weights be taken to be non-negative (and at least one positive) as an additional constraint to ensure that we obtain an average that falls inside the data range for the work we are presenting here. Such a choice is hardly "arbitrary". It simply constitutes an additional constraint that we choose to add, in addition to the assumed correlation structure, to obtain results that don't swing

outside the range of the input values.  We reject the reviewer's assertion that such an approach is incompatible with the use of this correlation model and that we are "fooling the math".  There are good reasons to do it.

Requiring each individual weight to remain non-negative, rather than the sum of the weights, can be thought of as a more local constraint than the global constraint imposed by positive definiteness.  In the text, we show that it may be used as a filter to throw out those scenes that cause problems locally.  This added constraint may be useful in guiding us to design a form for the error correlations that has the local boundedness constraint that we seek: the constant correlation model violates this local constraint generally, while the exponentially-decaying correlation model only requires about 5% of the data to be thrown out to satisfy it – perhaps if we keep looking we can find some other simple error correlation model that will satisfy both the local and global constraints on the weights.

We have added new discussion in the text explaining that the correlation models we assume should be expected to drive individual weights in our average into the negative range, there being no requirement against that in the math.   And we have added additional wording to the text explaining that we have forced the individual weights in the average to be non-negative in order to prevent the averages from falling outside the range of the values input to the average, a result that we feel is undesirable in our averages.

The sentence in question is this: "The resolution is limited both for computational reasons (the models must be run many dozens of times across the measurements to obtain the inverse estimate) and because the spatial coverage of the satellite measurements is currently not dense enough to resolve spatial scales much finer than this when solving at typical time scales (the gap in longitude between subsequent passes of a typical low-Earth- orbiting (LEO) satellite is ~25°, resulting in gaps of between 3° and 4° across a week, gaps which are generally never filled in further due to the repeat cycle of the satellite's orbit). "

The reviewer is correct that the longitude-separation calculation at the end of the sentence does not apply to GOSAT data taken over land, which may include data for 3, 5, or more data points taken in the cross-scan direction (or effectively 3, 5, or more parallel tracks of data per orbit groundtrack).  For GOSAT, the resulting gap size would have to be divided by 3, 5, or more, resulting in potentially a finer scale being resolvable over land.

To be clearer, we have added "taking a single swath of data along its orbit path" after "(LEO) satellite".

> l.43: why would it make little sense to assimilate the measurements
> individually? From the text, it is obvious that it is so much easier
> than trying averages. So, this can make a lot of sense. Personally,
> I would still prefer the averaging but for reasons that are not
> discussed here (numerical stability, but this is only a feeling).
The sentence in question: "The individual OCO-2 retrievals are generally averaged together along-track across some distance closer to the model grid box size before being assimilated in the inversion: this is because the modeled measurements to which the true measurements will be compared in the inversion are available only at the grid box resolution, so it makes little sense to assimilate each measurement individually when assimilating a coarse-resolution summary value will do just as well."

The answer is that it reduces the computational load related to the assimilation of the data by as much as a factor of 240 (the maximum number of individual scenes per 10-second span) if one averages the data beforehand. We don't understand why the reviewer feels that it would be easier to process each point individually in the assimilation, rather than averaging beforehand, unless he/she intends to ignore error correlations altogether in the process (and even then, the computational load would still be up to 240 times greater). If the same error correlations are to be considered in both cases, the effort required to do so is the same, whether done inside the code or outside, beforehand. But the computational savings obtained by pre-computing the averages remains, either way.

> l.48: interesting comment… The dependence of the correlations on
> the scene questions the representativeness of the ACT measurements
> used here. This key element is only briefly touched on in the warning in line 181.
The sentence spanning line 48 is this one: "$CO_2$ mixing ratios in the upper part of the atmospheric column (at all levels but the immediate surface layer) feel the influence of multiple flux locations at the surface due to atmospheric mixing, causing errors in adjacent measurements to be highly correlated there. "

It is not clear to us how the comment relates to this sentence. It is also not clear to us what argument the reviewer is trying to make here. If the comment implies that one cannot attempt to use a correlation model with a blanket correlation length applying across all data points, when local correlation lengths are higher or lower than that due to local conditions, then we must disagree. It is still better to try to use a more-accurate correlation model, even if the non-negativity constraint must be added as well, than not to attempt to account for error correlations at all.

> l.123: OCO-2 was already defined in l. 39. Same comment for ASCENDS a bit later.

We thank the reviewer for catching these -- we have removed the extraneous re-definitions.

> l.168: the motivation behind removing the linear trend is obscure.
> For the constant value removal, I do not see how this affects the
> calculation of the autocovariances.
Pertains to this sentence: "We then detrend this weighted difference timeseries across each flight leg – subtracting either a single constant value or a linear trend Y(x), where x is the along-track distance."

The reviewer is correct that a constant offset should not effect the spectrum returned by a harmonic analysis, insofar as the harmonic analysis usually solves for the average of the timeseries as part of the analysis such that it does not affect the harmonic terms (that is, the harmonic terms are solved for to describe the *variability* of the time series, which is described as deviations from the mean). This depends on whether one's particular routine does in fact solve for the mean and remove it or not, however. Our routine does not, so we are forced to pre-compute it and pre-subtract it before presenting the variability timeseries to the routine. Since the main factor that we address here is whether removing the linear trend or not makes a difference, we will refer to the case where only the constant offset (but not the linear trend) has been removed as the "linear trend not removed" case.

Linear trends are often also removed from a timeseries of data before performing a correlation analysis, in order to avoid including the harmonic terms needed to describe the trend in with the terms needed to describe the stationary variability. Not removing the trend in a harmonic analysis is equivalent to assuming that the trend repeats itself in a sawtooth-like pattern when the data span is repeated end-to-end. The spectrum needed to describe the trend is red: not removing the trend before doing a harmonic analysis will result in a spectrum in which the lower frequencies elements are exaggerated, making it more difficult to assess the longer frequency terms that pertain only to the stationary variability (i.e. the non-trend part).

To be conservative, we have chosen to use the spectrum computed with the trend not removed, giving the longer correlation length (20 km vs. 15 km) and resulting in less information being retained as the data are averaged across a given span length. We would also be comfortable with the more aggressive assumption that the trend should be removed, however (giving the shorter ~15 km length scale). We have chosen to show results for both cases to help illustrate the uncertainty in the calculation of the length scale.

Also, to be clearer, we have also replaced "Constant value removed" with "Linear trend not removed" in Figure 1.

> l.208 and 219: Kalman filters are used to control fluid state variables,
> but not boundary conditions such as surface fluxes. They are outside the
> scope of the discussion, unless the authors refer to simplifications like

> the Kalman smoother, but in this case the assimilation window covers
> periods much larger than the observation error correlations lengths which
> are discussed here.
On line 208: "Most estimation methods used in global atmospheric trace gas inversion
work (Bayesian synthesis inversions, Kalman filters, variational data assimilation)
combine measurement information in different timespans as:"
On line 219: "This assumption of uncorrelated errors between different timespans is
built into the derivations of these inverse methods explicitly, for example in the Kalman
filter, in which the dynamical errors related to propagating the measurement
information from time to time are assumed to be uncorrelated with the measurement
errors themselves."

The reviewer is incorrect in asserting that boundary conditions such as surface fluxes
cannot be included as control variables in Kalman filters.  For the $CO_2$ flux estimation
problem that we are dealing with here, one can envision a state vector that includes
both the current 3-D $CO_2$ field and surface fluxes in its state vector to be estimated --
new measurements would modify both the $CO_2$ field and the surface fluxes, the two of
which would be expected to be highly correlated.  This would not be as effective as
including multiple past fluxes in that state, as well (a situation that would lead to it
becoming effectively a fixed-lag Kalman smoother), but that does not mean that it
would be an illegitimate model.  So Kalman filters are certainly not outside the scope of
discussion.  And the Kalman smoother is not a simplification of the Kalman filter -- the
reverse is actually true (the filter is a simplification of the more general case of the
smoother, for only one lag step in the state).  It is not clear what point is being made
related to the assimilation window length.

> Section 3.1.3: Some main elements (the tridiagonal influence matrix, the statistically-
> optimal inflation factor) have been shown years ago by Chevallier et al
> (https://agupubs.onlinelibrary.wiley.com/doi/full/10.1029/2007GL030463)
> in a short paper.  It may have been too brief, but was much more accessible, I think.

Yes, we have been remiss in not mentioning this reference, given the good work that
Chevallier has done in characterizing errors in our flux inversion problem and given the
similar approach that he took in that reference to what we have used here (especially
the use of the tri-diagonal covariance inverse, but also the use of an inflation factor,
even if the precise form for computing it was not provided in that reference).  This was
an accidental oversight.  We hope to rectify that here by adding the following sentence
after equation (17):  "(Note that Chevallier et al. (2007) handled exponentially-decaying
correlated errors as well and used a similar tridiagonal matrix for the inverse of the
covariance.)"

Response to Comments from Reviewer #2:

> General comments.
> Authors propose an innovative approach to the construction of the
> error covariance matrixes for $X_{CO_2}$ observations by OCO-2 satellite.
> The method relies on comparison of OCO-2 retrievals to the collocated
> $X_{CO2}$ observations by an airborne lidar made along ground tracks of
> several tenth of km in length, and thus avoids use of model simulations
> for estimating the amplitude and spatial statistics of the OCO-2
> retrieval errors, which constitutes a major innovation presented in
> this work. Authors derive spatial error correlation lengths in the
> range of 10 to 20 km, which is a new result, based solely on
> observations. Authors consider 2 variants of the spatial error
> correlation models that are used to evaluate the errors and errors
> correlations for target 10 second averages derived from intermediate
> 2-second mean data, and propose ways to overcome related technical
> difficulties. The paper is well written and can be accepted after minor revisions.

> Detailed comments

> Although the problems and remaining deficiencies are extensively
> discussed in the Summary and conclusions, it is worth revisiting
> a couple of topics below:

>  The paper would benefit from deeper look at the problem with negative
> covariance elements, addressed in Sect. 3.2. Similar problem with
> spatial correlations would appear with covariance matrixes for
> surface fluxes errors in the same inverse problem for which the data
> uncertainty matrix is constructed. But the problem of negative elements
> for surface fluxes is not widely known, thus there is a possibility
> that the problem here is caused by an unrealistic design of the
> covariance matrix, but not by statistical properties of the data in hand.

Please see the response above to Reviewer #1's comments on lines 20 and 331. We argue there that we are justified in demanding non-negative weights for the average as an additional constraint we impose upon the problem to keep the average from falling outside the range of the input values

Reviewer #2 argues above that a similar negative weight problem ought to exist when handling correlated errors in the fluxes (as, for example, when averaging fluxes to a coarser time resolution, while properly accounting for the off-diagonal terms in the error covariance matrix), but this has never been described, suggesting that there should not be a problem with negative weights in specifying correlated data errors, either, and that some error has been made in the assumptions used in this paper, or in

the construction of the covariance matrix.  While that is not the case, this analogy has in fact been very helpful to us in adjusting our thinking, and we thank the reviewer for bringing it up.

Any covariance matrix with off-diagonal terms (including the ones we construct using our correlation models) can potentially have the elements of vector $\mathbf{R}^{-1}\mathbf{1}$ (the individual weights In our average, as we define then) assume negative values.  This would be true both of the measurement error covariance matrices that we deal with in this paper, as well as the flux error covariance matrices that the reviewer brings up in this comment. The reason for this is that there is no fundamental requirement in the form of the covariance matrices that requires each individual weight to stay non-negative.  Instead, the key criterion is that the *sum* of the weights of the terms going into the average must be greater than zero.  In that case, the denominator $\mathbf{w}^T\mathbf{1} = \mathbf{1}^T\mathbf{R}^{-1}\mathbf{1}$ in (5) and (6) always remains greater than zero and the weighted average is always well-defined. The sum of the weights, $\mathbf{w}^T\mathbf{1}$, stays positive whenever the covariance matrix, $\mathbf{R}$, is positive definite (since then $\mathbf{R}^{-1}$ is also positive definite, and therefore $\mathbf{1}^T\mathbf{R}^{-1}\mathbf{1} > 0$, by definition of positive definiteness).  Steps are always taken in estimation methods to ensure that covariance matrices remain positive definite, in the face of round-off error or noise driving them in that direction: divergence occurs in the Kalman filter, for example, when this condition is violated, and dynamic noise is added when propagating the covariance matrix forward in time to prevent this problem.  So there is does not need to be a problem in the design of our covariance matrix for this to happen – it is a natural feature of covariance matrices with off-diagonal terms.  And it (the existence of negative weights) ought to occur when dealing with flux error covariance matrices, as well, even if this is not generally well known.

There is no problem, as long as the user is satisfied with the net impact of the data (the mean data value in the case where the data is averaged before assimilating, or the net effect of the data in the inversion if each individual scene is assimilated separately) potentially falling outside the range of that given by the individual data when these correlation models are used.  However, we do see this as a problem – we do not want our average to fall outside the range of our input data -- and seek to mitigate it, while hopefully retaining the other advantages of the correlation models.  To do that, we must impose an additional, more stringent, constraint in demanding that each individual weight be non-negative, rather than just the sum of the weights.  This criterion (that each individual weight be non-negative) keeps the average value from straying outside the range of the data being averaged – this is certainly reasonable, and, since it is an effect that is caused by assuming a non-zero error correlation model itself in the first place, it is solved by adding an additional constraint outside of the correlation model. We have modified our text in Section 3.2 to discuss the points raised above.  We continue to demand the additional constraint that each individual weight be non-negative as before, but the additional text puts that decision into better perspective.

> Unfortunately authors do not provide information on the amplitude of

> the correlating component – what is a fraction of the (OCO-2 – lidar)
> difference that is correlated at 10 km or less scale. If that is only
> a fraction of the 1.5-2 ppm of error as found by comparison with TCCON,
> then the origin and spatial scales of remaining is unknown, thus it can
> be treated as random and uncorrelated. Bell et al (2020) notice that
> the correlations at local scale are pretty low, they write: "We conclude
> from these low correlations that for an average scene with no strong
> variability in the $X_{CO2}$ field, OCO-2 and the MFLL do not typically "see"
> the same small-scale features". In practice, the adopted level of
> correlation coefficients as shown on Eq. 15 do not appear justified by
> the comparison with MFLL and may possibly come from a separate source.

Yes, the coefficients shown in (15) come from a separate source – an independent analysis make by Dr. Susan Kulawik – we have added a note in the text that she did not use the MFLL data in her analysis, but rather based it in part on TCCON and *in situ* aircraft observations.  Actually, we feel that the +0.3 correlation coefficient calculated by Kulawik for use over land agrees quite well with the correlations that we computed using the MFLL data, shown in Figure 1, when applied to 10-second averages: the integral of +0.3 from 0 to 67.5 km gives a value of about 200, compared to an integral under the blue line on the left panel of Figure 1 of something very close to that across the same range on the x axis.  This agreement, based on very different data sources, gives us some confidence in the results of our MFLL-OCO2 analysis.

The reviewer is correct that we should have reported the magnitude of the variability corresponding to the correlations we report.  This can be computed directly from the spectrum shown in Figure 1 by multiplying the blue curve by the normalization factor that we have divided by to get the correlation spectrum.  For the case in which we remove the trend, that factor is 0.84 sigma when we analyze x_i / sigma_i, and 0.59 ppm when we analyze x_i itself without dividing by the associated uncertainty.  For the case in which we do not remove the trend, the values are 1.04 sigma and 1.003 ppm.  This means that the RMS difference between the MFLL and OCO-2 values is about 1.0 ppm for the non-detrended case, so that the correlation coefficient of +0.3 associated with separation distances of from 20 to 40 km is describing an RMS variability of about 1.04 * sqrt(0.3) = 0.57 ppm.  Or in terms of the fraction of the variability described at scales of 10 km or less that the reviewer asks about, anywhere from 100% of the variability at zero separation distance to about sqrt(0.37)=60% of the variability (in terms of concentration difference instead of its square) at a separation distance of 8 km (the first dot to the right of zero in Figure 1).  This is a substantial portion of the full variability, so we don't think it would be fair to discount it as being unimportant.  It is also not fair to discount it compared to the RMS difference of 1.5 to 2 ppm (OCO-2 compared to TCCON) since that number includes biases in both TCCON and OCO-2 that would drop out in any analysis of variability between the two.  The true RMS error in the OCO-2 data taken over land is down in the 1.2-1.5 ppm range at the moment, so the 1 ppm RMS difference between MFLL and OCO-2 is capturing about half of that – the

other half could plausibly be attributed to longer correlation scales than those that can be assessed here. We have added text discussing the magnitude of the variability associated with the MFLL – OCO-2 differences we have analyzed here.

> Minor comments, technical corrections:

> L9 'Errors in the $CO_2$ retrieval method have long been thought to be
> correlated at these fine scales' – It would be more accurate/safe to
> say that data are correlated, rather than the errors.

By data, we both mean retrievals here. We actually do mean to point to correlations in the errors rather than in the data themselves, since the errors are what are being quantified in the covariance matrices we discuss. Since the data are obviously strongly correlated due to their large background (~400 ppm) and large variability on seasonal and synoptic timescales, it might appear safer to only talk about them, but that is really besides the point. We know that the accuracy of the retrievals is strongly tied to the accuracy of the assumed surface properties, atmospheric scatterers, and atmospheric gases needed as part of the retrieval, and the *a priori* values assumed for these in the retrievals all have errors, so it is not surprising that we should talk about correlations between errors in the retrieved values themselves, rather than just in their values.

> L44 Alternatively, one can call this 'summary value' an 'average value'

Ok, we made that change.

> L48 Authors imply the errors here correspond to model-observation
> difference, and contributed by model errors related to smoothing due
> to coarse model resolution. Its better to define somewhere above this
> point what is implied by 'errors'.
The errors referred to here are measurement errors, more specifically errors between the retrieved and true $X_{CO2}$ values. We think the sentence on lines 45-46 already does a good job specifying the errors we are discussing: "Whether the individual OCO-2 retrievals or some coarser-resolution average measurement are assimilated into the inverse model, correlations between the errors in the individual $CO_2$ measurements must be considered." The following sentences delve into what factors cause errors in the $X_{CO2}$ retrievals, and some of these factors do involve modeled variables used in the retrievals. We do not get into the exact cause of those errors, because that is not our focus here.

> L86 Adding some MIP paper reference should be useful here (eg Crowell et al. 2019).

Yes it would – we have added a reference to Crowell et al. (2019) at the end of this long sentence.

> L167 Any rationale for detrending Y rather than X itself?

Analyzing the timeseries $Y_i = X_i / \text{sigma}_i$ instead of $X_i$ itself is better, because it places the proper weight on deviations that are large in a statistical sense (in sigma space) rather than in an absolute sense. If just $X_i$ was analyzed, it would be dominated by large deviations in places where the measurements are less certain, and the parts of the time series containing the most reliable data would be de-emphasized. That said, we also performed the analysis directly on the $X_i$ time series and the spectrum did not change much. A note on this has been added to the text.

> L203 As the MFLL data are first aggregated to 7-9 km blocks, it
> appears that one needs to clarify here on how the analysis would
> become useful for finer scales.

As noted here, because of this initial aggregation into 7-9 km blocks, scales finer than this cannot be addressed by this analysis. To be able to say something at finer scales, this blocking would have to be done across a shorter length scale -- this could be done down to the 2.3 km length of an individual OCO-2 cross-scan, at the expense of increased noise in the MFLL measurements. Whether the best fit to the spectrum would remain in the 15-20 km range determined here or would decrease would remain to be seen. A note to this effect has been added to the text.

> L221 Although temporally uncorrelated errors are convenient for
> Kalman filters, it does seem to be an excessive requirement, need to
> add a reference to appropriate text, if exists.
This sentence is being referred to: "This assumption of uncorrelated errors between different timespans is built into the derivations of these inverse methods explicitly, for example in the Kalman filter, in which the dynamical errors related to propagating the measurement information from time to time are assumed to be uncorrelated with the measurement errors themselves."

Kalman filters were designed originally for use inside real-time control loops, e.g., for a missile, rocket, or statically-unstable fighter aircraft. All the information needed to define the estimate at a given time is derived from new measurements at that time and the previous estimate propagated forward in time with a dynamical model. The errors between the different values in the new data vector may be correlated, and the dynamical errors may be correlated, but explicit correlations between the state estimates at different times can only be included by adding the state estimates at previous times into the state vector (a step that effectively turns the Kalman filter into a fixed lag Kalman smoother). We had added a reference to Applied Optimal Estimation, Gelb ed., The M.I.T. Press, 1974, 374 pp. to support this.

> L627 Mistype: correct 'Zendolo' to 'Zenodo'

Corrected, thanks for catching that.

> L660 For web document, need to give url.

Thanks for catching this.  We have added the link:
https://cce.nasa.gov/ascends_2015/ASCENDS_FinalDraft_4_27_15.pdf

---

## Author Response (AR1)

We'd like to thank the reviewers for their helpful comments.  Those addressing the issue of negative weights in our average and the fact that they are a natural consequence of our correlation models were particularly helpful to our understanding.  We have updated our text to reflect this updated viewpoint.

Responses to comments from Reviewer #1

> The authors study the correlations between errors in
> $X_{CO2}$ satellite retrievals, based on reference lidar
> measurements, and discuss various ways to account for
> them in atmospheric inversions. The paper looks a bit
> like the clean minutes of a brainstorming meeting:
> every sentence is well written but the logical flow is
> curvy and difficult to follow. The authors have not
> done enough to make their thoughts accessible and to
> take the text beyond elaborate speculation, perhaps
> simply because their thinking is not yet ripe for
> publication.

Where the reviewer has been able to give specific comments, we have attempted to address them (below).  If the reviewer can detail what in particular he/she finds inaccessible or mere speculation, we would be happy to address those points, as well.

Maybe it doesn't matter, as the paper
> will be cited anyway given the role of this activity
> for the OCO-2 team, but for the few who will bother
> to read it, it may be a daunting task, perhaps in the end wasted.

We hope our replies to the detailed comments below will address this reviewer's issues.

> I am listing a number of comments here to help clarify the presentation.

>   Footnote 1, p. 25: the disclaimer here is a bit hidden, but
> it is actually essential. Basically, if the "good reasons" listed
> here are correct, all results of the paper can be ignored. This
> observation could be fatal for the patient reader who painfully
> reaches this page… In the end, nothing is given to convince the
> reader that the MFLL-OCO-2 differences do indeed represent OCO-2
> errors, that the two scales of correlation lengths found (10 and
> 20 km) should be used at all in OCO-2 error models. It's embarrassing.

There is no cause for embarrassment here.  We have attempted to calculate a correlation length scale using the limited data that is available for that purpose.  When more data become available to test our conclusions, that should certainly be done -- and

if the reviewer can suggest some additional data that we could use to refine these estimates, we would be happy to work with it. In the meantime, the derivations presented here allowing the newly-calculated correlation length scale to be used remain valid regardless of what precise value is used for that quantity. So no, the reviewer's assertion that "all results of the paper can be ignored" if the MFLL correlation length calculation is not perfect is not correct at all. We feel that the reviewer is being overly negative in his/her assessment of the situation – we have merely added some cautions in the footnote to indicate what factors might affect the correlation length that we calculate.

> There is good and interesting math here, but the authors belittle
> it by arbitrarily rejecting certain math results: why should the
> negative weights not be physical (l. 20) or considered undesirable
> (l.331)? They simply follow from the authors' correlation model:
> if the authors are not satisfied with this consequence, they should
> change the model rather than fooling the math.

We thank this reviewer and Reviewer #2 for highlighting our treatment of the negative weight issue. We now agree with this reviewer that the negative weights that we obtain with both the constant and exponential-decay error correlation models are a natural, "physical", and expected consequence of these models. They only become potentially "undesirable" or "a problem" for the application we have chosen to look at: the calculation of averages from the data. This is because the usual definition of a weighted average specifies that the weights be non-negative (so that the average falls inside the range of the averaged values). Applications that do not involve averaging (for example, assimilation of the individual data values) would not have to deal with this issue explicitly. Also, if one chooses to take a broader view of what an average is, and to accept the negative weights and average values outside the range of the inputs, then yes, nothing more needs to be done, and the equations that we obtain using those correlated average assumptions (in Section 3.1) may be used straight, without the need for the additional data rejection criterion that we use to enforce the traditional definition of a weighted average. We will leave the discussion laying out approaches for enforcing the non-negative weight criterion (Section 3.2) in the document, however, to help those readers who prefer to obtain averages that follow the traditional definition of a weighted average. We investigate where the negative weight constraint is violated, why it is violated, and how this may be avoided (if desired). An advantage of investigating the negative weight issue is that this provides insight into the possibly-undesired effects of the assumed error correlations that may extend to applications other than averaging (e.g. the assimilation of individual, un-averaged, data values), which data cause these effects, and how these effects might be mitigated.

We do not agree with this reviewer that, if one does not like the consequences of these negative weights (the average obtained being outside of the range of the values averaged), one should therefore throw out the correlation model and find another one.

We feel that the approaches we have taken (either using an uncorrelated weighted average to calculate the mean, or using the correlated average in combination with a filtering approach to throw out those high-uncertainty, low-weight data points that cause the negative weights) are reasonable if one desires to obtain an average value inside the range of the input values.

Non-negative weights are often given as a basic requirement in defining a weighted average. Wikipedia, while not an unimpeachable academic reference, reflects broader practice when they give on their "weighted arithmetic mean" page the mathematical definition of such a mean as follows:
"Formally, the weighted mean of a non-empty finite multiset of data $\{ x_1, x_2, … , x_n \}$ , with corresponding *non-negative* weights $\{ w_1, w_2, … , w_n\}$ is
$<x> = sum\{w_i x_i\} / sum\{w_i\}$ ...
Therefore, data elements with a high weight contribute more to the weighted mean than do elements with a low weight. *The weights cannot be negative.* Some may be zero, but not all of them (since division by zero is not allowed)." [emphasis ours]

The practical effect of any individual weight being allowed to be negative is that the mean value computed may fall outside the range of the data going into the average – a result that is contrary to the very idea of an average. We are certainly justified in imposing the requirement that all individual weights be taken to be non-negative (and at least one positive) as an additional constraint to ensure that we obtain an average that falls inside the data range for the work we are presenting here. Such a choice is hardly "arbitrary". It simply requires an additional constraint that we may choose to add, in addition to the assumed correlation structure, to obtain averages that don't swing outside the range of the input values. We reject the reviewer's assertion that such an approach is incompatible with the use of this correlation model and that we are "fooling the math". There are good reasons to do it.

In the text, we show that requiring each individual weight to remain non-negative leads naturally to the use of a filter that throws out those scenes that cause the out-of-range averages. In the case of the constant correlation model, about half of the scenes violate the negative weight constraint, and it is not clear whether such a filter, implemented in an iterative fashion, would ever converge to a satisfactory subset of data points. In the case of the exponentially-decaying correlation model, on the other hand, only about 5% of the data need to be thrown out to satisfy the constraint, for the example we have used here. The use of the exponential-decay correlation model plus the negative weight filter thus seems like a reasonable approach to use with our data in order to reap the benefits of the correlated error model in calculating both the average and the uncertainty on it. Perhaps if we would have kept looking, we could have found some other relatively simple error correlation model that would not have resulted in negative weights occurring. While that is an interesting scientific question to investigate, it was not our goal in this paper – here, we have investigated two simple correlation models, one of which is well-suited to the way we have analyzed the MFLL

data, and we have found an approach (i.e. using the additional filter) that suits our purposes well.

We have added new discussion in the text explaining that the correlation models we assume should be expected to drive individual weights in our average into the negative range, there being no requirement against that in the math, and have also added a simple example of that in action.   And we have added additional wording to the text explaining that one may force the individual weights in the average to be non-negative in order to prevent the averages from falling outside the range of the values input to the average, if one feels that that is desirable for one's work, for example when one is explicitly computing averages of the data.  The locations of the changes in both the original and new (latexdiff-created) versions of the manuscript, are as follows:

(Line numbers in old / new (latexdiff) document:     Change)

20 / 20-22:  Replace "A small percentage of the data that cause non-physical negative averaging weights in the model are thrown out."
    with  "Considering correlated errors can cause the average value to fall outside the range of the values averaged: two strategies for preventing this are presented."

106-108 / 113-120:  Replace "Section 3.2 discusses a pathology in this approach for two of the error models – for some retrieval uncertainty value combinations, the weights given to some terms in the weighted average may go negative, violating a key criterion for any average – and gives a couple fallback options for handling it."
       with "When averaging data with unequal uncertainty values, considering correlated errors can cause the average value computed to fall outside the range of the input values to be averaged.  While this is a correct and natural consequence of the correlated error assumption, it does violate a key condition usually specified when defining a weighted average to prevent just that behavior: that all the weights be non-negative.  Section 3.2 discusses this issue in more detail and lays out a couple fallback options that we have used to stay with non-negative-weighted averages, while still reaping the benefits of the correlated error assumption."

324-325 / 346-347: Remove "However, both methods (treating the correlations in averages or in assimilating the data individually) may suffer a serious problem, described next. "

326 / 348: Replace "The problem of negative weights"
    with "Negative weights and their implications"

327-331 / 349-357: Replace "In defining any weighted average of the sort assumed in equation (5), the individual weights are required to be non-negative; some could be allowed to be zero, but not all of them, so that the denominator does not go to zero. It is unfortunately the case that negative weights may occur for both the constant and

exponential error correlation models that we have outlined in the previous two sections. The practical effect of negative weights is that the average value computed can fall outside of the range of the values input to the average, a feature generally not considered desirable in an average."

with "In the definition of a weighted average, the weights on the averaged values are usually required to be non-negative, with at least one weight being positive. Non-negative weights can cause the averaged value to fall outside the range of the values to be averaged, a result that is generally considered undesirable in an average, and the added requirement on the sign of the weights prevents this. However, under certain conditions the average values given by (13) and (28) can give negative weights and out-of-range average values."

332 / 358-359: Replace "This very out-of-range problem"
   with "This out-of-range behavior"
   Also, remove "recently".

340 / 366: Replace "The exponential correlation model suffers a similar problem."
   with "The exponential correlation model can also yield negative weights."

350 / 376-400: Replace "How can we sidestep this issue for practical problems? Could one try just discarding the retrievals with higher retrieval uncertainties $\sigma_j$?"
   with:
   "A simple example helps explain why the error correlation models drive the weights negative and the average value out of range of the input values. Consider two data points, each measuring a quantity for which the true value is $X_{true}=0$: let the value and uncertainty on these points be: $x_1=1$, $x_2=-1$, $\sigma_1= \sigma_2/\beta$. The error covariance matrix, $\mathbf{R}$, given by $\text{diag}(\sigma_1, \sigma_2)$ [1 c; c 1] $\text{diag}(\sigma_1, \sigma_2)$, describes the (correlated) errors: the differences between the measurements and the truth, which, since the truth equals zero, are just the values of the measurements themselves. $\mathbf{R}^{-1} = \text{diag}(\sigma_1^{-1}, \sigma_2^{-1})$ [1 -c; -c 1] $\text{diag}(\sigma_1^{-1}, \sigma_2^{-1}) / (1-c^2)$. Equation (6) gives the correlated mean: $[(\sigma_1^{-2} - \sigma_2^{-2})/(1-c^2)]/ [(\sigma_1^{-2} - 2c\ \sigma_1^{-1} \sigma_2^{-1} + \sigma_2^{-2})/(1-c^2)] = (\sigma_1^{-2} - \sigma_2^{-2})/ (\sigma_1^{-2} - 2c\ \sigma_1^{-1} \sigma_2^{-1} + \sigma_2^{-2}) = (\beta^2 - 1) / (\beta^2 - 2c\beta + 1)$. For $\beta=1$ (both uncertainties being the same), $X_{avg}=0$ for all values of the correlation coefficient, c, except $c=\pm1$. But for $\beta>1$, $X_{avg}$ moves more positive, closer to the measurement with more information or lower uncertainty, until $\beta=1/c$, at which point $X_{avg}=+1=x_1$. For larger $\beta$ values, $X_{avg}>1$, that is, outside the range of the two data values being averaged. Apparently, the correlated average, taking a clue from the value of the higher-uncertainty input, $x_2$, believes that the errors on both $x_2$ and (because of the positive correlation) $x_1$ are negative ($x_2$ being negative), and corrects for these errors by choosing a more positive value for the average value than the relative weighting of the two measurements, if uncorrelated, would otherwise require. When the difference between the uncertainties, $\beta$, is large enough (compared to $1/c$), the average value is driven outside the range of the input values. The weight on $x_2$ is driven negative to achieve this. For this error correlation model, all this makes sense.

If the correlated averages given by (13) and (28) are physically realistic, why not use them, even if they do not conform to the usual requirements of the weighted average? If it is clear that one's chosen model for the error correlations is correct, then yes, they should be used. But if one is not entirely sure of the model, that might be one reason to be hesitant to accept an average value that falls outside the range of the inputs. Is there an intermediate approach to fall back on that enforces the usual non-negative weight constraint for the average while still garnering the benefits of the correlated error models? One could try discarding the retrievals with higher retrieval uncertainties $\sigma_j$ that seem to be driving the weights negative."

402 / 453: change "problem" to "issue".

537 / 588-589:  Replace "As a workaround, the fallback model presented in Sect. 3.2.1 was used instead"
    with  "Because the team was uncertain how physically-realistic these average values were at that time, they fell back to using the model presented in Sect. 3.2.1 instead"

553-554 / 605-606:  Replace "We discovered that this model suffers from the same negative weight problem that our constant-correlation model did"
      with "We discovered that this model produced negative weights just as our constant-correlation model did"

> l.35-37: the sentence seems general, but does not apply to GOSAT in practice.
The sentence in question is this: "The resolution is limited both for computational reasons (the models must be run many dozens of times across the measurements to obtain the inverse estimate) and because the spatial coverage of the satellite measurements is currently not dense enough to resolve spatial scales much finer than this when solving at typical time scales (the gap in longitude between subsequent passes of a typical low-Earth-orbiting (LEO) satellite is ~25°, resulting in gaps of between 3° and 4° across a week, gaps which are generally never filled in further due to the repeat cycle of the satellite's orbit). "

The reviewer is correct that the longitude-separation calculation at the end of the sentence does not apply to GOSAT data taken over land, which may include data for 3, 5, or more data points taken in the cross-scan direction (or effectively 3, 5, or more parallel tracks of data per orbit ground track).  For GOSAT, the resulting gap size would have to be divided by 3, 5, or more, resulting in potentially a finer scale being resolvable over land.  However, for most of the GOSAT mission, a 3-point cross-scan pattern has been used, and, given the fact that this cross-scanning is not used over the oceans, and that clouds reduce the data availability from GOSAT (with a ~10 km field-of-view diameter) compared to OCO-2 (with a ~2-3 km FOV diameter), the difference between the data coverage from the two missions is less than it might otherwise seem.

However, to be clearer, we have added "taking a single thin swath of data along its orbit path" after "(LEO) satellite" on line 37 / 39.

> l.43: why would it make little sense to assimilate the measurements
> individually? From the text, it is obvious that it is so much easier
> than trying averages. So, this can make a lot of sense. Personally,
> I would still prefer the averaging but for reasons that are not
> discussed here (numerical stability, but this is only a feeling).
The sentence in question: "The individual OCO-2 retrievals are generally averaged together along-track across some distance closer to the model grid box size before being assimilated in the inversion: this is because the modeled measurements to which the true measurements will be compared in the inversion are available only at the grid box resolution, so it makes little sense to assimilate each measurement individually when assimilating a coarse-resolution summary value will do just as well."

The answer is that it reduces the computational load related to the assimilation of the data by as much as a factor of 240 (the maximum number of individual scenes per 10-second span) if one averages the data beforehand. We don't understand why the reviewer feels that it would be easier to process each point individually in the assimilation, rather than averaging beforehand, unless he/she intends to ignore error correlations altogether in the process (and even then, the computational load would still be up to 240 times greater). If the same error correlations are to be considered in both cases, the effort required to do so is the same, whether done inside the code or outside, beforehand. But the computational savings obtained by pre-computing the averages remains, either way.

> l.48: interesting comment... The dependence of the correlations on
> the scene questions the representativeness of the ACT measurements
> used here. This key element is only briefly touched on in the warning in line 181.
The sentence spanning line 48 is this one: "$CO_2$ mixing ratios in the upper part of the atmospheric column (at all levels but the immediate surface layer) feel the influence of multiple flux locations at the surface due to atmospheric mixing, causing errors in adjacent measurements to be highly correlated there. "

It is not clear to us how the comment relates to this sentence. It is also not clear to us what argument the reviewer is trying to make here. If the comment implies that one cannot attempt to use a correlation model with a blanket correlation length applying across all data points, when local correlation lengths are higher or lower than that due to local conditions, then we must disagree. It is still better to try to use a more-accurate correlation model than not to attempt to account for error correlations at all. Yes, one could attempt to construct a more complicated correlation model in which the correlations depend on the scene, but that would involve a level of complexity that we

did not intend to address in this paper (though we do specify a separate correlation length over land from that over the oceans).

> l.123: OCO-2 was already defined in l. 39. Same comment for ASCENDS a bit later.

We thank the reviewer for catching these -- we have removed the extraneous re-definitions, including one for MFLL (new lines 134-135, 149-152).

Pertains to this sentence: "We then detrend this weighted difference timeseries across each flight leg – subtracting either a single constant value or a linear trend Y(x), where x is the along-track distance."

The reviewer is correct that a constant offset should not effect the spectrum returned by a harmonic analysis, insofar as the harmonic analysis usually solves for and subtracts off the average of the timeseries as part of the analysis so that it does not affect the harmonic terms (that is, the harmonic terms are solved for to describe the *variability* of the time series, which is described as deviations from the mean). This depends on whether one's particular routine does in fact solve for the mean and remove it or not, however. Our routine does not, so we are forced to pre-compute it and pre-subtract it before presenting the variability timeseries to the routine. Since the main factor that we address here is whether removing the linear trend or not makes a difference, we will refer to the case where only the constant offset (but not the linear trend) has been removed as the "linear trend not removed" case.

Linear trends are often also removed from a timeseries of data before performing a correlation analysis, in order to avoid including the harmonic terms needed to describe the trend in with the terms needed to describe the stationary variability. Not removing the trend in a harmonic analysis is equivalent to assuming that the trend repeats itself in a sawtooth-like pattern when the data span is repeated end-to-end. The spectrum needed to describe the trend is red: not removing the trend before doing a harmonic analysis will result in a spectrum in which the lower frequency elements are exaggerated, making it more difficult to assess the longer frequency terms that pertain only to the stationary variability (i.e. the non-trend part).

To be conservative, we have chosen to use the spectrum computed with the trend not removed, which gives the longer correlation length (20 km vs. 15 km) and results in less information being retained as the data are averaged across a given span length. We would also be comfortable with the more aggressive assumption that the trend should be removed, however (which gives the shorter ~15 km length scale). We have chosen to show results for both cases to help illustrate the uncertainty in the calculation of the length scale.

Also, to be clearer, we have replaced "Constant value removed" with "Trend not removed" in Figure 1.

> l.208 and 219: Kalman filters are used to control fluid state variables,
> but not boundary conditions such as surface fluxes. They are outside the
> scope of the discussion, unless the authors refer to simplifications like
> the Kalman smoother, but in this case the assimilation window covers
> periods much larger than the observation error correlations lengths which
> are discussed here.

On line 208: "Most estimation methods used in global atmospheric trace gas inversion work (Bayesian synthesis inversions, Kalman filters, variational data assimilation) combine measurement information in different timespans as:"

On line 219: "This assumption of uncorrelated errors between different timespans is built into the derivations of these inverse methods explicitly, for example in the Kalman filter, in which the dynamical errors related to propagating the measurement information from time to time are assumed to be uncorrelated with the measurement errors themselves."

The reviewer is incorrect in asserting that boundary conditions such as surface fluxes cannot be included as control variables in Kalman filters. For the $CO_2$ flux estimation problem that we are dealing with here, one could envision a state vector that includes both the current 3-D $CO_2$ field and surface fluxes in its state vector to be estimated -- new measurements would modify both the $CO_2$ field and the surface fluxes, the two of which would be expected to be highly correlated. This would not be as effective as including multiple past fluxes in that state, as well (a situation that would lead to it becoming effectively a fixed-lag Kalman smoother), but that does not mean that it would be an illegitimate model. So Kalman filters are certainly not outside the scope of discussion. And the Kalman smoother is not a simplification of the Kalman filter -- the reverse is actually true (the filter is a simplification of the more general case of the smoother, for only one lag step in the state). It is not clear what point is being made related to the assimilation window length. If 'assimilation window' refers to the timespan of fluxes included in the state, then this could be as short as a couple time steps of the transport model or as long as many months, depending on the flux optimization timespan and the number of flux time steps included in the state vector – for shorter flux optimization spans, this could be getting close to important time correlation lengths in the problem.

> Section 3.1.3: Some main elements (the tridiagonal influence matrix, the statistically-
> optimal inflation factor) have been shown years ago by Chevallier et al
> (https://agupubs.onlinelibrary.wiley.com/doi/full/10.1029/2007GL030463)
> in a short paper. It may have been too brief, but was much more accessible, I think.

Indeed, we have been remiss in not mentioning this reference, given the good work that Chevallier has done in characterizing errors in our flux inversion problem and given the similar approach that he took in that reference to what we have used here (especially the use of the tri-diagonal covariance inverse, but also the use of an inflation factor, even if the precise form for computing it was not provided in that reference). This was an oversight. We hope to rectify that here by adding the following sentence after equation (17):

293 / 314-315: "(Note that Chevallier (2007) handled exponentially-decaying correlated errors as well and used a similar tridiagonal matrix for the inverse of the covariance.)"
as well as adding that source to the reference list.

Responses to comments from Reviewer #2:

> General comments.
> Authors propose an innovative approach to the construction of the
> error covariance matrixes for $X_{CO2}$ observations by OCO-2 satellite.
> The method relies on comparison of OCO-2 retrievals to the collocated
> $X_{CO2}$ observations by an airborne lidar made along ground tracks of
> several tenth of km in length, and thus avoids use of model simulations
> for estimating the amplitude and spatial statistics of the OCO-2
> retrieval errors, which constitutes a major innovation presented in
> this work. Authors derive spatial error correlation lengths in the
> range of 10 to 20 km, which is a new result, based solely on
> observations. Authors consider 2 variants of the spatial error
> correlation models that are used to evaluate the errors and errors
> correlations for target 10 second averages derived from intermediate
> 2-second mean data, and propose ways to overcome related technical
> difficulties. The paper is well written and can be accepted after minor revisions.

> Detailed comments

> Although the problems and remaining deficiencies are extensively
> discussed in the Summary and conclusions, it is worth revisiting
> a couple of topics below:

>  The paper would benefit from deeper look at the problem with negative
> covariance elements, addressed in Sect. 3.2. Similar problem with
> spatial correlations would appear with covariance matrixes for
> surface fluxes errors in the same inverse problem for which the data
> uncertainty matrix is constructed. But the problem of negative elements
> for surface fluxes is not widely known, thus there is a possibility
> that the problem here is caused by an unrealistic design of the
> covariance matrix, but not by statistical properties of the data in hand.

Please see the response above to Reviewer #1's comments on lines 20 and 331.
We argue there that one is justified in demanding non-negative weights for the average
as an additional constraint imposed upon the problem to keep the average from falling
outside the range of the input values. We agree that our correlation models do not
require this and that it is an additional constraint that one may choose to add; given our
focus on averaging in this manuscript, we do choose to add it, or at least to discuss it as
a possible approach to use.

Reviewer #2 argues above that a similar negative weight problem ought to exist when
handling correlated errors in the fluxes (as, for example, when averaging fluxes to a
coarser time resolution, while properly accounting for the off-diagonal terms in the

error covariance matrix), but this has never been described, suggesting that there should not be a problem with negative weights in specifying correlated data errors, either, and that some error has been made in the assumptions used in this paper, or in the construction of the covariance matrix.  While we believe that we have not made some such error in describing this effect, this analogy that the reviewer has put forward has in fact been very helpful to us in adjusting our thinking, and we thank the reviewer for bringing it up.

Any covariance matrix with off-diagonal terms (including the ones we construct using our correlation models) can potentially have the elements of vector $\mathbf{R}^{-1}\mathbf{1}$ (the individual weights In our average, as we define then) assume negative values.  This would be true both of the measurement error covariance matrices that we deal with in this paper, as well as the flux error covariance matrices that the reviewer brings up in this comment. The reason for this is that there is no fundamental requirement in the form of the covariance matrices that requires each individual weight to stay non-negative.  Instead, the key criterion is that the *sum* of the weights of the terms going into the average must be greater than zero.  In that case, the denominator $\mathbf{w}^T\mathbf{1} = \mathbf{1}^T\mathbf{R}^{-1}\mathbf{1}$ in (5) and (6) always remains greater than zero and the weighted average is always well-defined. The sum of the weights, $\mathbf{w}^T\mathbf{1}$, stays positive whenever the covariance matrix, $\mathbf{R}$, is positive definite (since then $\mathbf{R}^{-1}$ is also positive definite, and therefore $\mathbf{1}^T\mathbf{R}^{-1}\mathbf{1} > 0$, by definition of positive definiteness).  Steps are always taken in estimation methods to ensure that covariance matrices remain positive definite, in the face of round-off error or noise driving them in that direction: divergence occurs in the Kalman filter, for example, when this condition is violated, and dynamic noise is added when propagating the covariance matrix forward in time to prevent this problem.  So there does not need to be a problem in the design of our covariance matrix for this to happen – it is a natural feature of covariance matrices with non-zero off-diagonal terms.  And it (the existence of negative weights) ought to occur when dealing with flux error covariance matrices, as well, even if this is not generally well known.

   There is no problem, as long as the user is satisfied with the net impact of the data (the mean data value in the case where the data is averaged before assimilating, or the net effect of the data in the inversion if each individual scene is assimilated separately) potentially falling outside the range of that given by the individual data when these correlation models are used.  However, we do see this as a problem when we explicitly set about to compute an average: we do not want our average to fall outside the range of our input data and so we seek to mitigate it, while hopefully retaining the other advantages of the correlation models.  To do that, we must impose an additional, more stringent, constraint in demanding that each individual weight be non-negative.  This criterion (that each individual weight be non-negative) keeps the average value from straying outside the range of the data being averaged – this is certainly reasonable to require, and can be solved by adding an additional constraint outside of the correlation model.

We have modified our text in Section 3.2 to discuss the points raised above – please see our responses to Reviewer #1 above for the details of the changes we've made to the manuscript to address them.  We continue to demand the additional constraint that each individual weight be non-negative as before, but the additional text puts that decision into better perspective.

The question of why this "problem" of negative weights has never been discussed as it relates to flux correlations is an interesting one.  Modelers routinely average results at coarser time and space resolutions than they solve for in inversions (for example, averaging at the monthly and continental scales from results estimated originally at weekly scales across model grid boxes of 100s of km on a side), so why has this issue not been flagged before"?   Our first guess is that most researchers are not performing an average of the type done in (6), weighting with the inverse of the covariance matrix.  But for those who are, we suggest that they have not seen this effect because the scales that are averaged across are generally quite a bit larger than the scales at which the errors are correlated.  Consider the exponentially-decaying-with-separation-distance correlation model, for which the individual weights go negative (according to equation (36)) when $\sigma_i > <\sigma> \cosh(\Delta x/L)$, where L is the correlation length scale, $\Delta x$ the typical separation distance between the averaged quantities, and $<\sigma> = [(\sigma_{i+1}^{-1} + \sigma_{i-1}^{-1})/2]^{-1}$.  The weights-going-negative condition rarely occurs when $\cosh(\Delta x/L)$ is large, that is when $(\Delta x/L)$ is greater than 2 or 3.  For surface $CO_2$ fluxes, the strongest time correlations occur at the daily and synoptic scales, so for a global flux inversion estimating fluxes at weekly scales, $\Delta x/L$ is, in fact, about 7 for the daily frequency and about 2 for the synoptic frequency, and the impact of time correlations the averages should be small.  For interannual flux error correlations (e.g. errors between January fluxes in Year 2 being correlated with the January fluxes in Years 1 & 3) this argument fails – if such correlations exist (probably) and are of significant magnitude (not clear), they should be expected to have an impact in multi-year averages.  (We believe few, if any, inversion groups have tried to impose such correlations on the *a priori* fluxes in their modeling, which suggests that they are probably also not looking for the impacts of such correlations *a posteriori*.  If it has not come up as an issue in the literature, this may suggest that the magnitude of such interannually-correlated errors is not large).  In terms of spatial scales over land, Chevallier et al., GBC, (2012) found correlation length scales of 100-200 km using flux towers: this may, however, represent the finest scale that could be resolved with such a network, and if there were flux towers available every 1 km in a regular grid, for example, then the correlation length scale obtained may have been finer.  It is likely, then, that the typical $\Delta x$ used in global inversions (say 300 km) is quite a bit larger than the typical L value (under 100 km, say), such that $\Delta x/L>3$ and the negative weight problem should be negligible.  Over the oceans, for which longer error correlation length scales might be thought to occur (due to greater homogeneity of surface conditions and due to chemical buffering), this is probably not the case.   The fact that the fluxes over the oceans are typically an order of magnitude smaller than those over land would make it challenging to separately identify averaging errors due to out-of-range (negative weight) conditions in the face of spill-over errors

from the land regions, but it would be interesting to look for this effect, perhaps first in OSSE experiments.  In summary, we should expect to see examples of these negative-weight-related out-of-range averaging effects also when averaging fluxes: possibly these will become more evident as finer-scale flux inversions are done more frequently.

> Unfortunately authors do not provide information on the amplitude of
> the correlating component – what is a fraction of the (OCO-2 – lidar)
> difference that is correlated at 10 km or less scale. If that is only
> a fraction of the 1.5-2 ppm of error as found by comparison with TCCON,
> then the origin and spatial scales of remaining is unknown, thus it can
> be treated as random and uncorrelated. Bell et al (2020) notice that
> the correlations at local scale are pretty low, they write: "We conclude
> from these low correlations that for an average scene with no strong
> variability in the $X_{CO2}$ field, OCO-2 and the MFLL do not typically "see"
> the same small-scale features". In practice, the adopted level of
> correlation coefficients as shown on Eq. 15 do not appear justified by
> the comparison with MFLL and may possibly come from a separate source.

Yes, the coefficients shown in (15) come from a separate source – an independent analysis make by Dr. Susan Kulawik – we have added a note on line 284 / 303-304 in the text that she did not use the MFLL data in her analysis, but rather based it in part on TCCON and *in situ* aircraft observations.  Actually, we feel that the +0.3 correlation coefficient calculated by Kulawik for use over land agrees quite well with the correlations that we computed using the MFLL data, shown in Figure 1, when applied to 10-second averages: the integral of +0.3 from 0 to 67.5 km gives a value of about 200, compared to an integral under the blue line on the left panel of Figure 1 of something very close to that across the same range on the x axis.  This agreement, based on very different data sources, gives us some confidence in the results of our MFLL-OCO2 analysis.

The reviewer is correct that we should have reported the magnitude of the variability corresponding to the correlations we report.  This can be computed directly from the spectrum shown in Figure 1 by multiplying the blue curve by the normalization factor that we have divided by to get the correlation spectrum.  For the case in which we remove the trend, that factor is 0.84 sigma when we analyze $x_i / \sigma_i$, and 0.59 ppm when we analyze $x_i$ itself without dividing by the associated uncertainty.  For the case in which we do not remove the trend, the values are 1.04 sigma and 1.003 ppm.  This means that the RMS difference between the MFLL and OCO-2 values is about 1.0 ppm for the non-detrended case, so that the correlation coefficient of +0.3 associated with separation distances of from 20 to 40 km is describing an RMS variability of about 1.04 * sqrt(0.3) = 0.57 ppm.  Or in terms of the fraction of the variability described at scales of 10 km or less that the reviewer asks about, anywhere from 100% of the variability at zero separation distance to about sqrt(0.37)=60% of the variability (in terms of concentration

difference instead of its square) at a separation distance of 8 km (the first dot to the right of zero in Figure 1). This is a substantial portion of the full variability, so we don't think it would be fair to discount it as being unimportant and attempt to treat it as random and uncorrelated. It is also not fair to discount it compared to the RMS difference of 1.5 to 2 ppm (OCO-2 compared to TCCON) since that number includes biases in both TCCON and OCO-2 that would drop out in any analysis of variability between the two. The true RMS error in the OCO-2 data taken over land is down in the 1.2-1.5 ppm range at the moment, so the 1 ppm RMS difference between MFLL and OCO-2 is capturing about half of that – the other half could plausibly be attributed to longer correlation scales than those that can be assessed here. We have added the following text discussing the magnitude of the variability associated with the MFLL – OCO-2 differences on lines 191/ 204-209:

"The magnitude of the correlated variability is given by multiplying the square roots of the correlation coefficients shown on Figure 1 by the normalization factors obtained in computing them (0.59 ppm / 1.003 ppm when the trend is removed / not removed, or in terms of multiples of the uncertainty assumed on each MFLL-OCO2 difference, 0.84 / 1.04 $\sigma$). Moving over to 20 km on the x-axis, a coefficient of 0.25 translates into a magnitude of 0.5 x (0.59 / 1.003) = 0.3 / 0.5 ppm, large enough to be a significant fraction of the systematic errors in the OCO-2 data taken over land, which have been calculated to be about 0.6 ppm by Kulawik et al. (2019)."

> L9 'Errors in the $CO_2$ retrieval method have long been thought to be
> correlated at these fine scales' – It would be more accurate/safe to
> say that data are correlated, rather than the errors.

(By data, we both mean retrievals here.) We actually do mean to point to correlations in the errors rather than in the data themselves, since the errors are what are being quantified in the covariance matrices we discuss. Since the data are obviously strongly correlated due to their large background (~400 ppm) and large variability on seasonal and synoptic timescales, it might appear safer to only talk about them, but that is really besides the point. We know that the accuracy of the retrievals is strongly tied to the accuracy of the assumed surface properties, atmospheric scatterers, and atmospheric gases needed as part of the retrieval, and the *a priori* values assumed for these in the retrievals all have errors, so it is not surprising that we should talk about correlations between errors in the retrieved values themselves, rather than just in their values.

> L44 Alternatively, one can call this 'summary value' an 'average value'

We changed "summary value" to "average that summarizes those values", on lines 44 / 46-47.

> L48 Authors imply the errors here correspond to model-observation
> difference, and contributed by model errors related to smoothing due
> to coarse model resolution. Its better to define somewhere above this
> point what is implied by 'errors'.

There are two sentences here that deal with errors, one ending on line 48 and one beginning there, and we are not sure from this comment which one the reviewer is referring to. The second sentence deals with errors in the parameters used in the radiative transfer modeling used in the retrievals. While the errors in these parameters are partly due to the coarse-resolution models from which some of these parameters are taken (e.g. the temperature profile taken from a reanalysis product), they are also due to model errors unrelated to resolution (e.g. errors in aerosol concentration, size, or type in the aerosol product used in the retrieval, or in the scattering model used with those aerosols). The errors discussed in the first sentence (the one ending on line 48) are less clearly described, and are probably what the reviewer is pointing to. The idea that errors in the $CO_2$ mixing ratios in the upper part of the column must be correlated, because the upper part of the column is affected by $CO_2$ fluxes that occurred far afield in both time and space, could be described more clearly. It comes from a model of the truth that assumes $CO_2$ mixing ratios in the interior of the atmosphere are determined only by surface fluxes and atmospheric mixing, and neglects any sources or sinks in the interior of the atmosphere. By the time surface flux perturbations to $CO_2$ have been transported up to the upper part of the column, they have generally be blown a good distance away from where they were emitted and have mixed with air emitted in surrounding gridcells. Any errors in the emitted fluxes (or in transport) will lead to difficulties in attributing the errors to flux errors in any one local area, due to the mixing blurring out the relationship. An error in upper level $CO_2$ then maps onto errors in surface fluxes that are highly correlated in space and time. Such a model could be used for the $CO_2$ prior used in the retrieval or as a measurement model relating measured or retrieved $CO_2$ mixing ratios to the surface $CO_2$ fluxes that cause them. Here it is better to think in terms of the latter, since the $CO_2$ priors used in the retrieval are not really built up from underlying fluxes explicitly. We have reworded the first two sentences in the paragraph to hopefully clarify what errors we are referring to:

45-48/ 51-55: Replace "Whether the individual OCO-2 retrievals or some coarser-resolution average measurement are assimilated into the inverse model, correlations between the errors in the individual $CO_2$ measurements must be considered. $CO_2$ mixing ratios in the upper part of the atmospheric column (at all levels but the immediate surface layer) feel the influence of multiple flux locations at the surface due to atmospheric mixing, causing errors in adjacent measurements to be highly correlated there."

with "Whether the individual OCO-2 $X_{CO2}$ retrievals or some coarser-resolution average of them are assimilated into the inverse model, correlations between the errors in the individual $X_{CO2}$ measurements must be considered. $CO_2$ mixing ratios in the upper part

of the atmospheric column (at all levels but the immediate surface layer) feel the influence of multiple flux locations at the surface due to atmospheric mixing, which widens and homogenizes the zone of influence as time goes on.   $X_{CO2}$ is a measure of $CO_2$ across the full column and is dominated by such effects: any error in $X_{CO2}$ will be translated into highly-correlated errors in neighboring surface fluxes when used as a measurement in an inversion model; similarly, any error in surface $CO_2$ flux in a forward model will result in highly-correlated errors in neighboring $X_{CO2}$ measurements influenced by these fluxes."

> L86 Adding some MIP paper reference should be useful here (eg Crowell et al. 2019).

Yes, we agree – we have added references to both Crowell *et al*. (2019) and Peiro *et al*. (2021) at the end of the first part of this sentence, before the hyphen, on line 88 / 95, and have updated them in the list of references:

Crowell, S., D. Baker, A. Schuh, S. Basu, A.R. Jacobson, F. Chevallier, J. Liu, F. Deng, L. Feng, K. McKain, A. Chatterjee , J.B. Miller, B.B. Stephens, A. Eldering, D. Crisp, D. Schimel, R. Nassar, C.W. O'Dell, T. Oda, C. Sweeney, P.I. Palmer, and D.B.A. Jones, The 2015-2016 carbon cycle as seen from OCO-2 and the global *in situ* network, *Atmos. Chem. Phys*., 19, 9797–9831, https://doi.org/10.5194/acp-19-9797-2019, 2019.

Peiro, H., Crowell, S., Schuh, A., Baker, D. F., O'Dell, C., Jacobson, A. R., Chevallier, F., Liu, J., Eldering, A., Crisp, D., Deng, F., Weir, B., Basu, S., Johnson, M. S., Philip, S., and Baker, I.: Four years of global carbon cycle observed from OCO-2 version 9 and *in situ* data, and comparison to OCO-2 v7, *Atmos. Chem. Phys*. *Discuss*., https://doi.org/10.5194/acp-2021-373, in review, 2021.

> L167 Any rationale for detrending Y rather than X itself?

Analyzing the timeseries $Y_i = X_i$ / sigma$_i$ instead of $X_i$ itself is better, because it places the proper weight on deviations that are large in a statistical sense (in sigma space) rather than in an absolute sense (the assimilation itself is essentially operating on the weighted measurements, not the unweighted ones).  If just $X_i$ was analyzed, it would be dominated by large deviations in places where the measurements are less certain, and the parts of the time series containing the most reliable data would be de-emphasized. Since this is true when applied to trends in the data, as well, we have chosen to detrend $Y_i$ preferentially.  That said, we have also performed the analysis by detrending the $X_i$ time series directly and the spectrum did not change much, as was already discussed on lines 173-177 / 185-186 of the manuscript.

> L203 As the MFLL data are first aggregated to 7-9 km blocks, it
> appears that one needs to clarify here on how the analysis would
> become useful for finer scales.

On lines 202-203 of the original manuscript, when we wrote "…in the process we will

get insight into how to handle data assimilated at finer scales, even on an individual retrieval-by-retrieval basis", we were thinking that the equations we obtained would provide insight across a wide range of scales on either side of the 10-second averaging span that we chose for our application. It is true, though, that because we have not examined the MFLL-OCO2 differences at a binning finer than 7-9 km (we could do so down to the 2.3 km along-track extent of the OCO-2 cross-scan), we cannot be sure what the correlation would look like were we to do so – maybe the correlation length scale would turn out to be finer than the 10 km length scale we plot in Figures 2 & 3, for example. We have chosen not to reassess the data at these finer scales, since this is not important for the main thrust of our paper. Instead, we have tried to make our original point clearer by rewording as follows:

202-203 / 221-222: Replace "but in the process we will get insight into how to handle data assimilated at finer scales, even on an individual retrieval-by-retrieval basis."

   with "but in the process we will get insight into how to handle data assimilated at coarser scales and at finer scales down to the 7-9 km MFLL binning size used here."

> L221 Although temporally uncorrelated errors are convenient for
> Kalman filters, it does seem to be an excessive requirement, need to
> add a reference to appropriate text, if exists.
This sentence is being referred to: "This assumption of uncorrelated errors between different timespans is built into the derivations of these inverse methods explicitly, for example in the Kalman filter, in which the dynamical errors related to propagating the measurement information from time to time are assumed to be uncorrelated with the measurement errors themselves."

We agree that this assumption may seem to be excessive when dealing with data that are clearly time correlated, such as the satellite data we address here, and we attempt to remove this assumption in this paper. One could, instead, derive versions of the popular estimation methods that account for such correlated errors explicitly. However, this assumption (that errors between $\mathbf{x}_1$ and $\mathbf{x}_2$ in (3) are uncorrelated) is built into the standard forms of many estimation methods, including the Kalman filter. In the standard form of the Kalman filter, the state estimate update step (in which measurement information is incorporated into the state) takes the form of equation (3) if $\mathbf{x}_2$ is taken to be the previous estimate of the state propagated to the current time step and $\mathbf{x}_1$ the new measurement information transformed into state space. Then (4) and (3) become $P_+^{-1} = P_1^{-1} + P_2^{-1} = H^T R^{-1} H + P_-^{-1}$ and $P_+^{-1} \mathbf{x}_+ = H^T R^{-1} \mathbf{z} + P_-^{-1} \mathbf{x}_-$, which can be put into the standard form of the Kalman filter update equations with some algebraic manipulation [here, the '-' and '+' notation indicates the state vector and its covariance matrix before and after the measurement update]. To obtain this standard form of the equations, the errors in $\mathbf{x}_1$ and $\mathbf{x}_2$ must be assumed to be uncorrelated, that is $E[d\mathbf{x}_1 \, d\mathbf{x}_2^T] = \mathbf{0}$. This is noted in derivations of the standard Kalman filter, for example equation

(7.2-3) on page 138 of Catlin (1989) and equation (4.2-11) in Gelb (1974), which have now been added to the text as references supporting this assertion (line 221 / 240).

Catlin, D.E., Estimation, Control, and the Discrete Kalman Filter, Springer-Verlag, New York, 1989, 274 pp.

Gelb, A., *ed*., Applied Optimal Estimation, The M.I.T. Press, Cambridge, MA, 1974, 374 pp.

> L627 Mistype: correct 'Zendolo' to 'Zenodo'

Corrected in two places, 604 / 656  &  627 / 679 -- thanks for catching that.

> L660 For web document, need to give url.

Thanks for catching this.  We have added the link:
https://cce.nasa.gov/ascends_2015/ASCENDS_FinalDraft_4_27_15.pdf
on line 660 / 718.  We have also added a reference to the ASCENDS final report (Kawa et al., 2018) on the same line.